# OMNISHOW: Unifying Multimodal Conditions for Human-Object Interaction Video Generation

**Donghao Zhou** [1][*]    **Guisheng Liu** [2][*]    **Hao Yang** [2]    **Jiatong Li** [2][†]    **Jingyu Lin** [3]    **Xiaohu Huang** [4]    **Yichen Liu** [2]
**Xin Gao** [2]    **Cunjian Chen** [3]    **Shilei Wen** [2][‡]    **Chi-Wing Fu** [1]    **Pheng-Ann Heng** [1][‡]

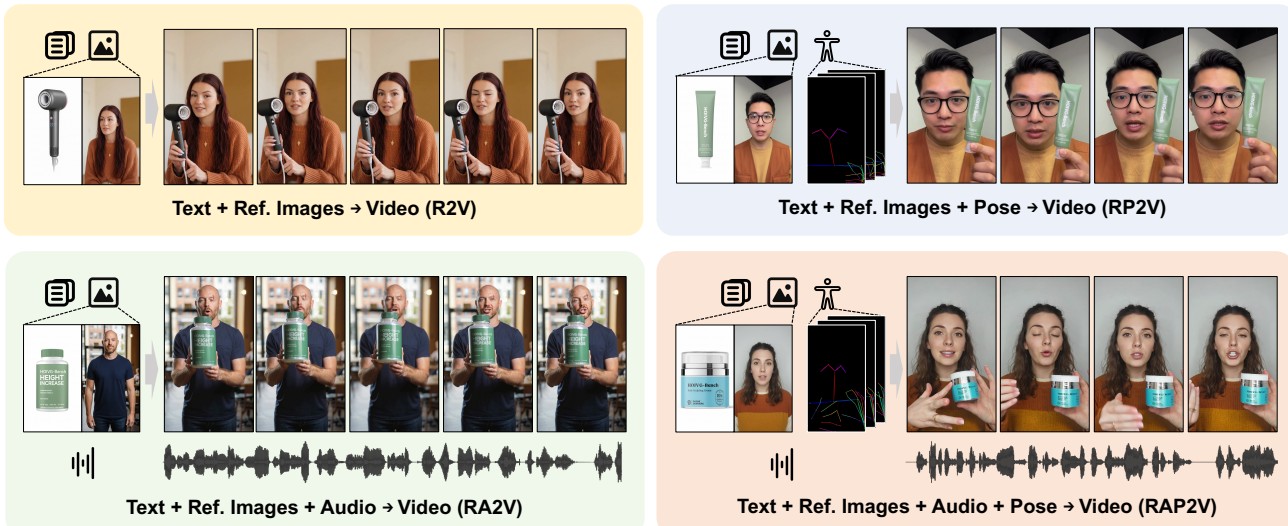

*Figure 1.* **Overview of OMNISHOW.** Our model unifies text, reference image, audio, and pose conditions to synthesize high-quality videos. It exhibits exceptional flexibility across diverse conditioning settings, from R2V, RA2V, RP2V to the most challenging RAP2V, consistently delivering industry-grade visual quality and precise multimodal alignment. *More immersive and diverse video demonstrations are available on our project page:* *https://correr-zhou.github.io/OmniShow/.*

## Abstract

In this work, we study **Human-Object Interaction Video Generation (HOIVG)**, which aims to synthesize high-quality human-object interaction videos conditioned on text, reference images, audio, and pose. We present **OMNISHOW**, the first all-in-one model tailored for this practical yet challenging task, capable of harmonizing multimodal conditions and delivering industry-grade performance. To overcome the trade-off between controllability and quality, we introduce **Unified Channel-wise Conditioning** for efficient image and pose injection, and **Gated Local-Context Attention** to ensure precise audio-visual synchronization. To effectively address data scarcity, we develop a **Decoupled-Then-Joint Training** strategy that leverages a multi-stage training process with model merging to efficiently harness heterogeneous sub-task datasets. Furthermore, to fill the evaluation gap in this field, we establish **HOIVG-Bench**, a dedicated and comprehensive benchmark for HOIVG. Extensive experiments demonstrate that OMNISHOW achieves overall state-of-the-art performance across various multimodal conditioning settings, setting a solid standard for the emerging HOIVG task.

[*]Equal contribution [†]Project lead [‡]Corresponding author [1]The Chinese University of Hong Kong [2]ByteDance [3]Monash University [4]The University of Hong Kong. Correspondence to: Shilei Wen <zhengmin.666@bytedance.com>, Pheng-Ann Heng <pheng@cse.cuhk.edu.hk>.

*Proceedings of the 43rd International Conference on Machine Learning*, Seoul, South Korea. PMLR 306, 2026. Copyright 2026 by the author(s).

## 1. Introduction

The rapid evolution of text-to-video generation models (Wan et al., 2025; Zhang et al., 2025; Wu et al., 2025; Chen et al., 2025b) has revolutionized content creation, especially in human-centric scenarios. While human-centric video gen-

eration has achieved impressive visual fidelity, real-world applications—such as e-commerce demonstrations, short video production, and interactive entertainment—demand precise controllability over specific subjects and their dynamics. We term this practical yet challenging task as **Human-Object Interaction Video Generation (HOIVG)**. Specifically, it requires synthesizing high-quality videos conditioned on four distinct inputs: a *text* prompt for global semantics, *reference images* for specific character and object appearance, *audio* for synchronized lip and body movements, and a *pose* sequence for explicit motion control.

Despite the progress, existing methods struggle to unify the diverse multimodal conditions required for HOIVG. Reference-to-Video (R2V) approaches (Liu et al., 2025b; Fei et al., 2025) focus on subject preservation but lack audio responsiveness, resulting in "silent" interactions. Conversely, Audio-to-Video (A2V) methods (Cui et al., 2025b; Gan et al., 2025b; Kong et al., 2025) aim for audio synchronization but only support an initial frame rather than reference images, limiting their applicability. While existing works (Hu et al., 2025; Chen et al., 2025a) attempt to combine audio and reference images, they overlook pose conditions, which are critical for realizing complex interactions that text cannot specify. Moreover, these methods are not tailored for HOIVG, often falling short of high-fidelity subject preservation required by this task. Although some approaches target this task (Xu et al., 2026; Wang et al., 2025c; Hu et al., 2025), they rely on mandatory inputs such as predefined object masks or trajectories and lack audio-driven abilities. Furthermore, recent joint audio-video generation models (Huang et al., 2025c; Low et al., 2025; HaCohen et al., 2026) rely on text prompts, offering limited customization capabilities. Consequently, there is no unified framework capable of harmonizing text, reference image, audio, and pose conditions in an end-to-end manner, all of which are required by the HOIVG task.

Achieving this unification presents three primary challenges: (i) A critical trade-off exists between multimodal controllability and generation quality. Naively introducing aggressive modifications to handle multimodal inputs typically disrupts the base model's pretrained generative priors. (ii) High-quality HOI datasets containing paired quintuplet data (conditions and the target video) are scarce. Instead, readily available resources are fragmented into isolated sub-tasks (*e.g.*, separate A2V datasets). (iii) The community currently lacks a dedicated and comprehensive benchmark to evaluate HOIVG under such diverse multimodal conditions, hindering sustainable research exploration in this field.

In this work, we propose **OMNISHOW**, an end-to-end framework designed for HOIVG (Figure 2). By simultaneously orchestrating text, reference image, audio, and pose conditions, OMNISHOW unlocks the full potential of multimodal

control while achieving superior generation quality (Figure 1). Our methodology is driven by a triad of efforts: *integrating efficient and pragmatic techniques, synergistically utilizing heterogeneous training data, and establishing a standardized benchmark*.

First, we introduce **Unified Channel-wise Conditioning** (Figure 2a). We augment the noisy video tokens by padding additional tokens (termed "pseudo-frame tokens") along the temporal dimension. Both the pose video and reference images are encoded by VAE, and then injected via the same channel-concatenation strategy: the pose tokens are concatenated with the noisy video tokens, and the image tokens with the pseudo-frame tokens. Furthermore, we employ a reconstruction loss on pseudo-frames to encourage the preservation of reference images' semantic details. This mechanism maintains the native input structure and token distribution of the base model, thereby minimizing the adaptation gap and achieving efficient, seamless injection.

Second, we design **Gated Local-Context Attention** (Figure 2b). An audio context packing strategy is leveraged to aggregate rich audio features that encapsulate sufficient contextual information. To achieve temporal alignment, we employ a masked attention mechanism that restricts video tokens to interact only with their corresponding audio segments. Furthermore, we incorporate learnable gating vectors to modulate the injection, which stabilizes early training and serves as an explicit indicator for the impact of audio cues. Collectively, these design choices facilitate precise audio-visual synchronization in generated videos.

Third, we develop **Decoupled-Then-Joint Training** (Figure 2c). We start by establishing multiple data pipelines to collect, filter, and organize diverse training data for HOIVG. To fully leverage these heterogeneous data, we initially train separate A2V and R2V models, allowing each to specialize in its respective modality. These models are then fused for joint training to harmonize text, reference images, and audio. Next, pose is introduced in the final stage to prevent over-reliance on this strong signal. Such a strategy not only boosts data utility but also validates that fusing specialized models benefits robust multimodal training.

For a systematic evaluation, we propose **HOIVG-Bench**, a dedicated and comprehensive benchmark for HOIVG. Experiments on this benchmark show that OMNISHOW delivers superior or competitive performance against existing R2V, A2V, and other methods in multiple multimodal control settings. We believe that OMNISHOW will set a solid standard for the emerging HOIVG task, and the methodological and empirical insights presented herein could inspire future advancements in video generation.

Our main contributions are summarized as follows:

- We propose OMNISHOW, the first-of-its-kind frame-

work for HOIVG, capable of harmonizing multimodal conditions. To achieve this, we introduce *Unified Channel-wise Conditioning* and *Gated Local-Context Attention*, which enable precise controllability without compromising generation quality.

- We develop a *Decoupled-Then-Joint Training* strategy, which leverages a multi-stage training process with model merging to efficiently harness heterogeneous data from diverse sub-task datasets, effectively circumventing the data scarcity of the HOIVG training.

- We establish *HOIVG-Bench*, a dedicated and comprehensive benchmark for HOIVG evaluation. Extensive experiments on HOIVG-Bench validate that OMNISHOW achieves state-of-the-art performance in various multimodal conditioning settings.

## 2. Related Work

**Controllable Video Generation** aims to synthesize videos conditioned on diverse inputs beyond text. Recent advances in visual generative models have motivated extensive research, from conditional image synthesis (Zhou et al., 2026; Lin et al., 2025b; Liu et al., 2026; Zhou et al., 2024; Song et al., 2025b; Qin et al., 2026; Chen et al., 2026; Huang et al., 2025a) to controllable video generation (Huang et al., 2025b; 2024a; Song et al., 2025a; Wang et al., 2025b; Ling et al., 2026; Wang et al., 2025a; Shao et al., 2025). Reference-to-Video (R2V) generation, also referred to as video customization, focuses on preserving the subject identity of input reference images (Yuan et al., 2025b; Liu et al., 2025b; Fei et al., 2025; Zhou et al., 2025). Driven by the demand for digital avatars, Audio-to-Video (A2V) generation has evolved from talking heads (Zhang et al., 2023b; Jiang et al., 2024) to portrait animation (Cui et al., 2025a; Lin et al., 2025a; Yi et al., 2025; Wei et al., 2025) and multi-person conversation (Zhong et al., 2025; Wang et al., 2025e; Kong et al., 2025). Concurrently, pose-guided approaches leverage explicit structural signals, ranging from skeleton maps (Xue et al., 2024a; Gan et al., 2025a) to dense correspondences (Xu et al., 2024), to direct the generation of human motion. Recently, there is a growing trend towards integrating multiple conditions (Hu et al., 2025; Jiang et al., 2025; Chen et al., 2025a; Kling Team et al., 2025). However, establishing a robust framework that collaborates text, reference image, audio, and pose conditions, which are required by our studied task, remains a significant open challenge.

**Human-Object Interaction Video Generation (HOIVG)** focuses on synthesizing realistic and vivid HOI videos grounded in multimodal conditions. Prior research on HOI has spanned in 3D reconstruction (Xu et al., 2021; Ye et al., 2023) and motion sequence synthesis (Diller & Dai, 2024; Peng et al., 2025). With the advancement of visual generative models, previous works have begun to investigate

HOI in image generation (Xue et al., 2024b; Chen et al., 2024; Fan et al., 2025). Recently, HOIVG has emerged as a prominent topic, driven by its critical role in real-world applications. AnchorCrafter (Xu et al., 2026) utilizes body skeletons, hand meshes, and object depth maps to guide the interaction; HunyuanVideo-HOMA (Huang et al., 2025d) proposes to use sparse human poses and object trajectory dots; and DreamActor-H1 (Wang et al., 2025c) rely on body mesh templates and object bounding boxes. However, these methods are all constrained by strict input requirements and cannot utilize audio cues, struggling to achieve satisfactory flexibility and generation quality. In contrast, our OMNISHOW supports flexible configurations of multimodal conditions, while delivering outstanding performance.

## 3. Methodology

As illustrated in Figure 2, our OMNISHOW generates high-quality videos by conditioning on a combination of multimodal inputs. Built upon Waver 1.0 (Zhang et al., 2025) (Section 3.1), a 12B MMDiT-based model, the framework comprises four key components: *Unified Channel-wise Conditioning* (Section 3.2) for efficiently injecting reference images and pose cues without disrupting the generative priors; *Gated Local-Context Attention* (Section 3.3) for ensuring precise synchronization between audios and human dynamics; *Decoupled-Then-Joint Training* (Section 3.4) for effectively harnessing heterogeneous datasets; and *HOIVG-Bench* (Section 3.5) for providing a comprehensive evaluation suite to systematically assess this task.

### 3.1. Preliminary

Waver 1.0 follows the latent diffusion paradigm, utilizing Wan 2.1 VAE (Wan et al., 2025) to compress video features as latent tokens. In the latent space, Flow Matching (Lipman et al., 2022) is adopted for training, where the objective is to minimize the discrepancy between the predicted velocity field $v_\theta$ and the ground-truth flow velocity $u$:

$$\mathcal{L}_{\text{FM}} = \mathbb{E}_{t,\mathbf{x}_0,(\mathbf{x}_1,\mathbf{e})} \left[ \|v_\theta(t, \mathbf{x}_{\text{in}}, \mathbf{e}) - u(\mathbf{x}_t|\mathbf{x}_1)\|^2 \right], \quad (1)$$

where $t$ represents the timestep, $\mathbf{x}_0 \sim \mathcal{N}(0, \mathbf{I})$ denotes the Gaussian noise, and $\mathbf{x}_1$ is the clean video tokens with $\mathbf{e}$ as the paired text embedding. The noisy video tokens $\mathbf{x}_t = \mathbb{R}^{N \times D}$ contains $N$ tokens with channel dimension $D$, and $u(\mathbf{x}_t|\mathbf{x}_1) = \mathbf{x}_1 - \mathbf{x}_0$ is the corresponding flow velocity.

Moreover, Waver 1.0 is a task-unified model supporting both the Text-to-Video (T2V) and Image-to-Video (I2V) tasks. To achieve this, it expands $\mathbf{x}_t$ via channel concatenation to form the model input:

$$\mathbf{x}_{\text{in}} = \text{Concat}(\mathbf{x}_t, \mathbf{c}, \mathbf{m}), \quad (2)$$

where $\mathbf{c} \in \mathbb{R}^{N \times D}$ represent the condition tokens, and $\mathbf{m} \in [0, 1]^{N \times 4}$ is the binary mask indicating the condition status.

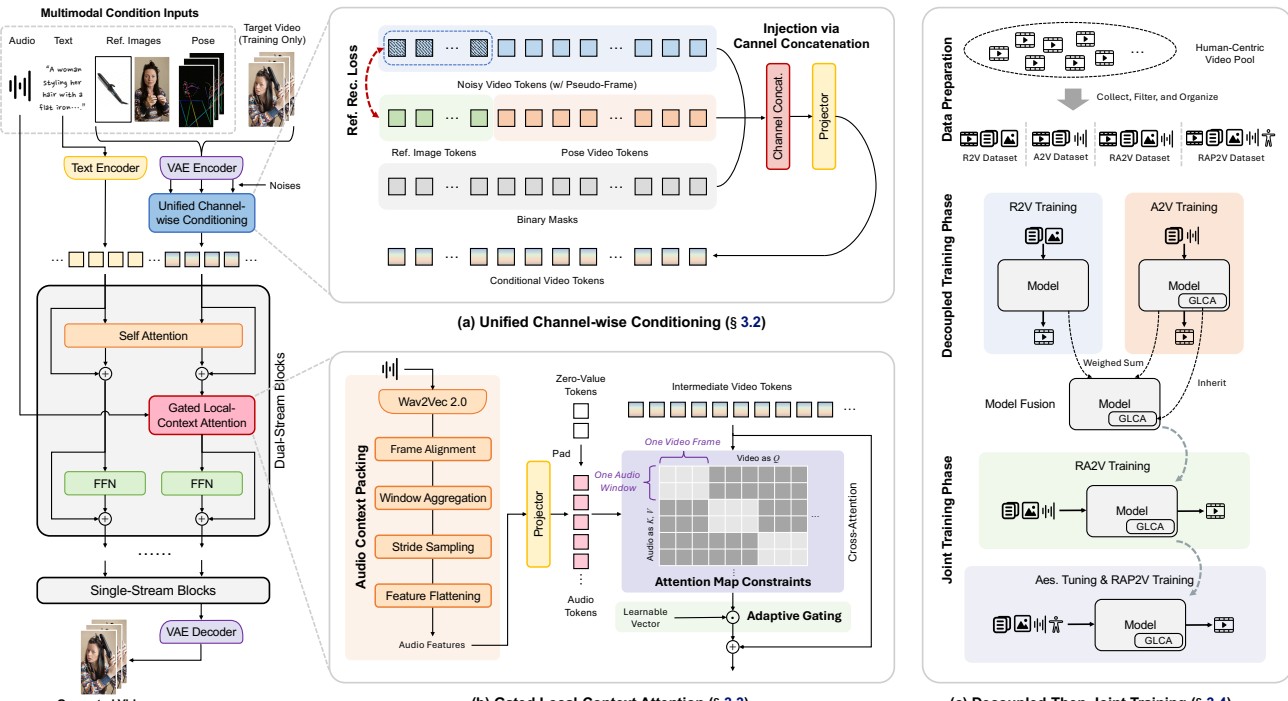

*Figure 2.* **Pipeline of OMNISHOW.** Our framework consists of: *(a) Unified Channel-wise Conditioning* (Section 3.2) effectively injects reference images and pose cues via unified channel concatenation. It augments noisy video tokens with pseudo-frames, which are supervised by a reference reconstruction loss to preserve semantic details. *(b) Gated Local-Context Attention* (Section 3.3) ensures precise audio-visual synchronization. It packs raw audio features with sufficient contextual information and injects them via masked attention to align video frames with corresponding audio segments, followed by adaptive gating to stabilize early training. *(c) Decoupled-Then-Joint Training* (Section 3.4) makes the efficient utilization of heterogeneous datasets possible. We first train specialized R2V and A2V models on separate sub-task datasets, then fuse them via weight interpolation, followed by joint fine-tuning to unify multimodal capabilities. Note that we omit some components, such as normalization, within the MMDiT blocks for brevity.

Specifically, for the T2V task, $\mathbf{c}$ is filled by black image tokens, and $\mathbf{m}$ is set to all zeros. As for the I2V task, to condition on the first frame, the corresponding part of $\mathbf{c}$ is replaced by the input first-frame image tokens, and the corresponding entries in $\mathbf{m}$ are set to 1.

### 3.2. Unified Channel-wise Conditioning

**Injection via Channel Concatenation.** To harmonize the injection of reference images and pose while maximally preserving the generation quality, we follow and extend the native conditioning paradigm. First, the pose sequence is rendered as an RGB video, and pose video tokens $\mathbf{p} \in \mathbb{R}^{N \times D}$, reference image tokens $\mathbf{r} \in \mathbb{R}^{N' \times D}$ are obtained by VAE encoding. Originally, $\mathbf{x}_t$ only provides $N$ conditioning slots. To structurally accommodate the joint injection, we expand this capacity by augmenting $\mathbf{x}_t$ with $N'$ pseudo-frame tokens $\mathbf{x}' \in \mathbb{R}^{N' \times D}$. Consequently, the conditions are injected via a unified channel-wise concatenation strategy:

$$\mathbf{x}_{\text{in}} = \text{Concat}([\mathbf{x}' \parallel \mathbf{x}_t], [\mathbf{r} \parallel \mathbf{p}], [\mathbf{m}' \parallel \mathbf{m}]), \quad (3)$$

where $\parallel$ denotes concatenation along the temporal axis, and $\mathbf{m}' \in \mathbb{R}^{N' \times D}$ is the corresponding augmented mask. This

unified design allows the model to efficiently assimilate the global appearance reference with temporally aligned pose cues simultaneously.

**Reference Reconstruction Loss.** Leaving pseudo-frame tokens $\mathbf{x}'$ as all-zero tensors provides no informative guidance for the model. To improve this, we initialize $\mathbf{x}'$ with the noisy reference image tokens perturbed by the same timestep $t$, and enforce a Flow Matching loss $\mathcal{L}_{\text{FM-ref}}$ to facilitate the reconstruction of reference images, with the loss weight set to 1. This strategy ensures synchronized denoising dynamics and explicitly compels the model to better perceive and faithfully retain high-fidelity semantic details, thereby ensuring more robust visual consistency with the conditional reference images.

**Advantage Analysis.** While the token concatenation strategy has proven effective in similar tasks (Tan et al., 2025; Ju et al., 2025), our method offers superior performance for task-unified video generation models (Figure 3 & Table 3a). We attribute this to *the minimization of the task adaptation gap*: instead of introducing hybrid tokens that bring substantial learning costs, we preserve the native input structure for conditioning, allowing the model to transfer the pre-

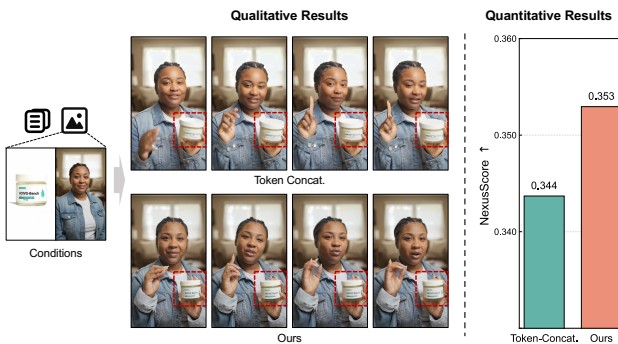

Figure 3. **Comparison of conditioning methods**, including a qualitative example and quantitative results based on the HOIVG-Bench. The proposed mechanism accurately preserves the visual appearance of the reference subject.

trained I2V capability efficiently. Such a design highlights the potential of advanced channel-wise conditioning as a worth-exploring solution to controllable video generation.

### 3.3. Gated Local-Context Attention

**Audio Context Packing.** We adopt an effective scheme to consolidate rich audio features from raw audio signals. First, the audio is fed into Wav2Vec 2.0 (Baevski et al., 2020), where representations from multiple layers are merged to capture both semantic and rhythmic attributes. Then, linear interpolation is employed to match the fps of the original video. To aggregate temporal context, we adopt a sliding window strategy with a size of $w = 5$, stacking neighbors for each audio feature along an extra dimension. These features are then sampled with a stride of $s = 4$ to align with the VAE temporal compression. Finally, the contextual features are flattened in chronological order, yielding dense 2D features rich in contextual information.

**Attention Map Constraints.** The packed features are processed by a shared audio projector and then interact with video tokens through cross-attention. To achieve precise audio-visual temporal alignment, we constrain each latent frame's video tokens to attend only with its corresponding $w$ audio tokens, resulting in a masked attention mechanism:

$$\text{Attn}(Q, K, V, M) = \text{softmax}\left(\frac{QK^T}{\sqrt{d_k}} + \log M\right) V, \quad (4)$$

where queries $Q$ are derived from video tokens, while keys $K$ and values $V$ come from audio tokens. $M$ is a binary matrix, where an entry of 1 allows interaction and 0 prevents it (here $\log 0$ is implemented as a large negative constant in practice). Notably, we zero-pad audio tokens to align with additional pseudo-frame tokens, a trick essential for accommodating reference image injection. By eliminating interference from irrelevant audio segments, this attention design enforces fine-grained correspondence between modalities, significantly enhancing audio-visual synchronization.

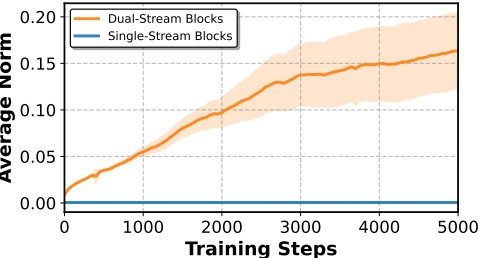

Figure 4. **Variation of the average norm of g**, which reflects audio impact in different blocks.

**Adaptive Gating.** Directly inserting newly initialized modules can disrupt pretrained feature distributions during early training (Zhang et al., 2023a). To avoid this, we introduce a learnable gating vector $\mathbf{g} \in \mathbb{R}^H$, initialized to a near-zero value of $1e - 5$, where $H$ is the token hidden dimension. With the modulation of $\mathbf{g}$, audio injection is formulated as:

$$\mathbf{h}_o = \mathbf{h}_i + \mathcal{F}_{\text{Attn}}(\mathbf{h}_i, \mathbf{a}) \odot \mathbf{g}, \quad (5)$$

where $\mathbf{h}_i, \mathbf{h}_o \in \mathbb{R}^{(N'+N) \times H}$ denote input and output video tokens, $\mathbf{a} \in \mathbb{R}^{N_a \times H}$ is audio tokens with token number $N_a$, $\mathcal{F}_{\text{Attn}}$ represents the complete attention operation including Equation (4) and associated projections, and $\odot$ is element-wise multiplication. We argue that this method offers distinct advantages over simply zero-initializing weights within $\mathcal{F}$. Beyond ensuring training stability, $\mathbf{g}$ also acts as an explicit indicator to guide our architectural design via revealing the magnitude of audio impact. Specifically, motivated by the empirical observation of $\mathbf{g}$ (Figure 4), we insert audio attention only into the dual-stream blocks for efficient injection. This strategic placement merely increases the model scale by ~2.5% (totaling 12.3B), while existing methods like HuMo (Chen et al., 2025a) increase model parameters by ~21.4% (totaling 17B).

### 3.4. Decoupled-Then-Joint Training

**Training Data.** The effectiveness of video generation relies on training data. However, high-quality HOIVG data is scarce, as a video requires valid text, reference images, audio, and pose conditions; failure in any condition leads to discarding the sample. Thus, we propose to collect heterogeneous data that meets specific sub-task standards for training. We first collect a massive human-centric video pool. Then, we build multiple pipelines to construct R2V, A2V, and Reference+Audio-to-Video (RA2V) datasets. Consequently, we curate a high-quality Reference+Audio+Pose-to-Video (RAP2V) subset for final fine-tuning. More details of data collection are included in Section A.

**Decoupled Training Phase.** We start by training specialized R2V and A2V models, utilizing their dedicated datasets (Figure 2c). Note that, for R2V training, we discard audio modules to retain the same architecture as the base model;

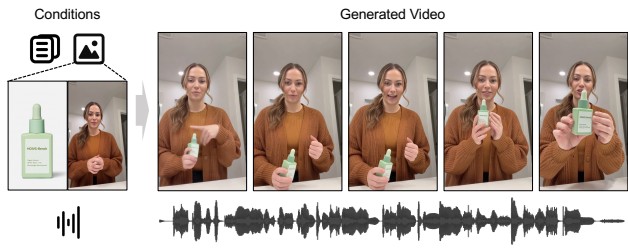

*Figure 5.* **Zero-shot RA2V generation via model merging.** Despite not being explicitly trained for this joint task yet, the merged model successfully generates videos that respect both reference images and audio inputs.

for A2V training, we follow the common paradigm to incorporate the first-frame image as an additional condition. Such a phase allows for the efficient utilization of heterogeneous sub-task data. Furthermore, it enables each model to master its specific modality conditioning, ensuring robust alignment before subsequent joint training.

**Joint Training Phase.** Subsequently, we merge the models by inheriting audio modules from the A2V model and linearly interpolating the rest (Figure 2c), with 0.6, 0.4 for the A2V and R2V models, respectively. The ratio selection is guided by a principled observation: audio synchronization (relying on fine-grained temporal alignment) is significantly more sensitive to weight disruption than visual identity (relying on global appearance features). Remarkably, *the merged model exhibits emergent RA2V capabilities* (Figure 5), evidencing the effectiveness of our strategy. It is then trained on the full RA2V dataset, followed by further fine-tuning on a high-quality subset to enhance naturalness and aesthetics. Notably, as a strong supervision signal, pose is introduced only in the final fine-tuning stage to prevent overfitting. Through this phase, we ultimately achieve a unified model capable of multimodal conditioning.

### 3.5. HOIVG-Bench

To systematically evaluate the capabilities of HOIVG under diverse multimodal conditions, we present **HOIVG-Bench**. Existing benchmarks often focus on limited-modality control (*e.g.*, text+pose or text+images), lacking a comprehensive assessment of the synergy between text, images (both human and object), audio, and pose signals, which are required by the HOIVG setting. HOIVG-Bench aims to bridge this gap by providing an evaluation suite comprising 135 carefully curated samples and dedicated metrics.

**Sample Construction.** In HOIVG-Bench, each sample contains a detailed caption, a human reference image, an object reference image, semantically aligned audio, and a coherent pose sequence. We construct these samples through a dedicated multi-stage pipeline: first curating raw human-object interaction videos from an in-house library, then deriving privacy-preserving human references and high-

quality object references, extracting pose sequences from the source videos, and finally synthesizing semantically matched speech audio conditioned on the target sample. We additionally perform manual quality control to remove samples with noticeable artifacts or excessive "AI-ness". Detailed sample construction procedures and design choices are provided in Section B.

**Evaluation Metrics.** The metrics are across five key dimensions: (1) *Text Alignment*: we employ VideoReward (Liu et al., 2025a) to predict the text alignment (TA) score; (2) *Reference Consistency*: we follow OpenS2V (Yuan et al., 2025a) to report FaceSim and NexusScore; (3) *Pose Accuracy*: we use average keypoint distances (AKD) and percentage of correct keypoints (PCK) (with an error threshold of 5%) based on DWPose (Yang et al., 2023); (4) *Audio-Visual Synchronization*: we adopt the Sync-C and Sync-D scores (Chung & Zisserman, 2016); and (5) *Video Quality*: we use VBench (Huang et al., 2024b) for the AES and IQA metrics, plus VideoReward (Liu et al., 2025a) for overall visual (VQ) and motion quality (MQ). Although our model supports video generation of up to 10 seconds, to ensure a fair comparison with baseline methods that only support short-clip generation, all quantitative metrics and qualitative analyses on HOIVG-Bench are standardized to 5-second video clips at 720p resolution in portrait mode.

## 4. Experiments

### 4.1. Setups

**Implementation Details.** We initialize our model from Waver 1.0 (Zhang et al., 2025). Training spans two resolutions of 480p and 720p. Unless otherwise specified, the large-scale training uses a computing cluster of 128 GPUs, each with 80GB RAM. To ensure computational efficiency, we employ Fully Sharded Data Parallel (FSDP) (Zhao et al., 2023) combined with the Ulysses-style sequence parallelism (Jacobs et al., 2023), utilizing BF16 mixed precision. The model is optimized by AdamW (Loshchilov & Hutter, 2017) with a learning rate of $3 \times 10^{-5}$ and a weight decay of 0.01. More details are provided in Section C.

**Compared Methods.** We compare OMNISHOW (12.3B) with leading open-source methods, including HunyuanCustom (13B) (Hu et al., 2025), HuMo (17B/1.7B) (Chen et al., 2025a), VACE (14B) (Jiang et al., 2025), Phantom (14B/1.3B) (Liu et al., 2025b), and AnchorCrafter (1.5B) (Xu et al., 2026). To ensure a thorough comparison, we evaluate several available variants with different model sizes for these baselines. However, as these methods do not support the full spectrum of four multimodal inputs inherent to OMNISHOW, the experiments are conducted across varying input settings (*i.e.*, R2V, RA2V, and RP2V) to align with the different capabilities of each baseline.

*Table 1.* **Quantitative comparison.** We compare our OMNISHOW with existing state-of-the-art methods, showing that OMNISHOW can achieve superior/competitive performance across diverse multimodal conditioning settings. The best and second-best results in each setting are marked in **bold** and underlined. The symbol "-" indicates that the metric is not applicable in the corresponding setting.

| Method | Text Align. | Reference Consistency | | Audio-Visual Sync. | | Pose Accuracy | | Video Quality | | | |
|---|---|---|---|---|---|---|---|---|---|---|---|
| | TA↑ | FaceSim↑ | NexusScore↑ | Sync-C↑ | Sync-D↓ | AKD↓ | PCK↑ | AES↑ | IQA↑ | VQ↑ | MQ↑ |
| *Text+Reference-to-Video (R2V)* | | | | | | | | | | | |
| HunyuanCustom (Hu et al., 2025) | 7.523 | 0.440 | 0.359 | - | - | - | - | 0.452 | 0.697 | 10.11 | 5.286 |
| HuMo-1.7B (Chen et al., 2025a) | 7.087 | 0.647 | 0.333 | - | - | - | - | 0.441 | 0.723 | 9.76 | 3.406 |
| HuMo-17B (Chen et al., 2025a) | 7.949 | 0.843 | 0.346 | - | - | - | - | 0.448 | 0.726 | 9.97 | 3.685 |
| VACE (Jiang et al., 2025) | 8.413 | 0.759 | 0.368 | - | - | - | - | 0.457 | 0.722 | 10.72 | 5.442 |
| Phantom-1.3B (Liu et al., 2025b) | 8.342 | 0.708 | 0.351 | - | - | - | - | 0.459 | 0.722 | 10.90 | 5.637 |
| Phantom-14B (Liu et al., 2025b) | **8.609** | **0.876** | 0.366 | - | - | - | - | 0.449 | **0.741** | 10.93 | 5.517 |
| OMNISHOW (Ours) | 7.746 | 0.874 | **0.389** | - | - | - | - | **0.468** | 0.740 | **11.12** | **5.885** |
| *Text+Reference+Audio-to-Video (RA2V)* | | | | | | | | | | | |
| HunyuanCustom (Hu et al., 2025) | 7.289 | 0.457 | 0.350 | 6.072 | 10.08 | - | - | 0.439 | 0.715 | 9.15 | 3.658 |
| HuMo-1.7B (Chen et al., 2025a) | 7.489 | 0.575 | 0.329 | 7.234 | 9.117 | - | - | 0.428 | 0.731 | 9.97 | 4.182 |
| HuMo-17B (Chen et al., 2025a) | **8.146** | 0.805 | 0.344 | 8.013 | 8.316 | - | - | 0.439 | 0.739 | 10.27 | 4.269 |
| OMNISHOW (Ours) | 8.093 | **0.810** | **0.369** | **8.612** | **7.608** | - | - | **0.465** | **0.742** | **10.86** | **5.554** |
| *Text+Reference+Pose-to-Video (RP2V)* | | | | | | | | | | | |
| AnchorCrafter (Xu et al., 2026) | 2.669 | 0.404 | 0.215 | - | - | 0.229 | 0.176 | **0.499** | 0.673 | 8.95 | 4.241 |
| VACE (Jiang et al., 2025) | **7.690** | **0.600** | 0.352 | - | - | 0.206 | 0.336 | 0.450 | 0.712 | 10.14 | **5.393** |
| OMNISHOW (Ours) | 6.526 | 0.474 | **0.418** | - | - | **0.174** | **0.460** | 0.447 | **0.722** | 10.28 | 4.937 |

## 4.2. Main Results

**Quantitative Comparison.** The quantitative results across three key settings are presented in Table 1. Our OMNISHOW demonstrates robust performance, achieving state-of-the-art or highly competitive results across the majority of metrics. Moreover, among 10B-scale models that meet industry-grade standards, OmniShow is the smallest and the most parameter-efficient. In the R2V setting, our method matches the reference preservation capabilities of specialized methods like Phantom-14B, as evidenced by our comparable FaceSim and NexusScore. Notably, OMNISHOW exhibits distinct advantages as a unified framework in more complex scenarios. In the RA2V setting, while dedicated baselines like HuMo-17B may show slight gains in TA, our approach delivers leading performance in other metrics like NexusScore and Sync-C. Moving to the RP2V setting, although the precise pose adherence introduces viewpoint shifts and facial morphology changes that affect FaceSim, OMNISHOW can still maintain robust video quality. These results underscore its ability to effectively orchestrate multimodal conditions within a cohesive architecture. It is worth noting that OMNISHOW is the only model supporting RAP2V generation, and there are no directly comparable counterparts. To further verify its capabilities, we compare OMNISHOW with a cascaded baseline in Section D.

**Qualitative Comparison.** The qualitative results in Figure 6 highlight the superiority of our method. Across varied scenarios, OMNISHOW exhibits high-fidelity reference preservation, natural motion dynamics, and precise audio-visual synchronization. In the R2V setting, unlike baselines that often rigidly paste objects onto human subjects at implausible sizes, our model ensures both visual fidelity and realistic composition. For the RA2V setting, OMNISHOW generates natural body movements alongside precise lip synchronization, avoiding the "overreaction" and "frozen body" issue common in previous methods. In the RP2V setting involving complex motion, our model demonstrates robustness in handling complex spatial interactions and large pose variations, accurately generating hand contacts and object appearance, whereas VACE struggles with following the pose and AnchorCrafter fails to preserve object identity. Since no existing methods simultaneously support all four conditions as ours do, a direct comparison in this full setting (*i.e.*, RAP2V) is absent. Moreover, we showcase more qualitative results across all four multimodal conditioning settings in Figure 1 and Section F, which further verify its effectiveness in unifying all modalities to generate coherent, expressive, and realistic videos.

**Human Evaluation.** To measure human preference, we conduct side-by-side comparisons in both the RA2V and RP2V settings, where user experience is paramount. Specifically, we engage a diverse demographic pool of evaluators, including 30 participants for the RA2V task and 33 participants for the RP2V task, to assess randomly selected subsets of 20 samples. As shown in Figure 7, the user studies show that human evaluators favor our results even when some objective scores are comparable. Moreover, our OMNISHOW is preferred in the majority of cases, demonstrating superi-

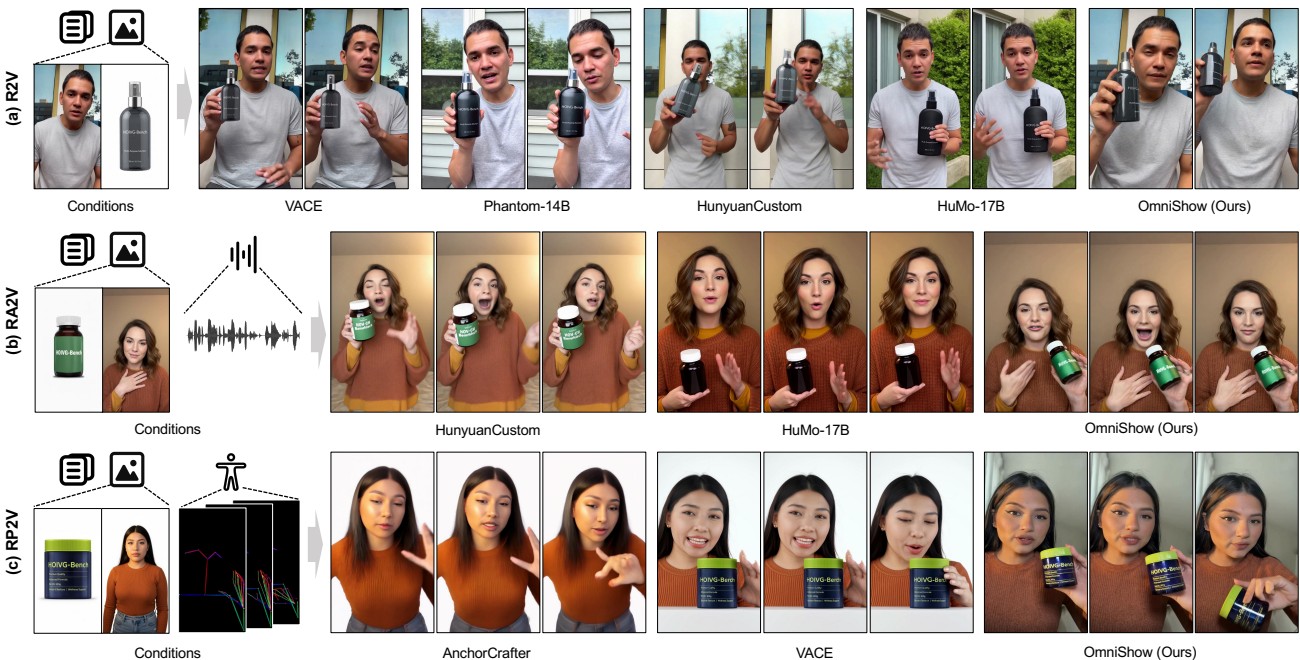

*Figure 6.* **Qualitative comparison.** We present generated results from our OMNISHOW and other methods across various multimodal condition settings. Note that text prompts are omitted for brevity.

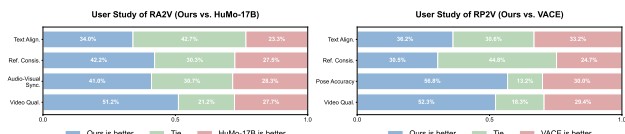

*Figure 7.* **Side-by-side human evaluation.** Judged by human evaluators, our OMNISHOW achieves superior performance in overall quality, showing great alignment with human preference.

*Table 2.* **Performance on the EMTD benchmark.** Note that metrics marked with "*" adopt the definitions from this benchmark, thus leading to different numerical ranges.

| Method | *IQA↑ | *AES↑ | Sync-C↑ | Sync-D↓ |
|---|---|---|---|---|
| FantasyTalking (Wang et al., 2025d) | 2.11 | 1.12 | 1.11 | 12.88 |
| HunyuanVideo-Avatar (Chen et al., 2025c) | 1.76 | 1.18 | 4.89 | 9.37 |
| Hallo3 (Cui et al., 2025b) | **2.31** | 1.48 | 4.26 | 10.22 |
| MultiTalk (Kong et al., 2025) | 2.07 | 1.30 | 6.34 | **8.47** |
| OmniAvatar (Gan et al., 2025b) | 2.16 | 1.31 | 5.40 | 9.13 |
| OMNISHOW-A2V (Ours) | 2.26 | **1.51** | **6.49** | 8.97 |

ority in both condition adherence and overall visual quality. We attribute this to smoother temporal dynamics and richer visual details in our generated videos, which are critical for perceived realism but might be overlooked by frame-level metrics. These evaluation results further highlight the superiority of our OMNISHOW.

### 4.3. Ablation Studies and Analysis

In this section, we present a series of experiments to evaluate the effectiveness of OMNISHOW, which include (i) comprehensive ablation studies on the proposed techniques, (ii) an additional evaluation on the A2V task, and (iii) an exploration of our model's broader applications. Note that unless otherwise stated, all runs, including those for our methods, are conducted on 8 GPUs and ensure fairness.

**Ablation of Unified Channel-wise Conditioning.** We first compare this visual injection mechanism against the token concatenation approach (Tan et al., 2025) in the R2V task. Results in Table 3a show our method yields superior video quality and reference consistency (also see Figure 3). This

highlights that adhering to the native conditioning structure is more efficient for visual injection in task-unified video generation models. Additionally, removing the reference reconstruction loss leads to a drop in visual fidelity, especially for human identity, validating its role in enforcing semantic preservation during training.

**Ablation of Gated Local-Context Attention.** We evaluate variants by removing audio context, attention map constraints, or adaptive gating in the A2V task. Results in Table 3b indicate that incorporating audio context is crucial for capturing temporal coherence, reflected in improved Sync-D, while attention map constraints significantly boosts synchronization by ensuring frame-wise interaction. Furthermore, disabling adaptive gating degrades final visual quality, confirming the necessity of audio feature modulation for training stability.

**Ablation of Decoupled-Then-Joint Training.** We compare our training strategy against conventional ones us-

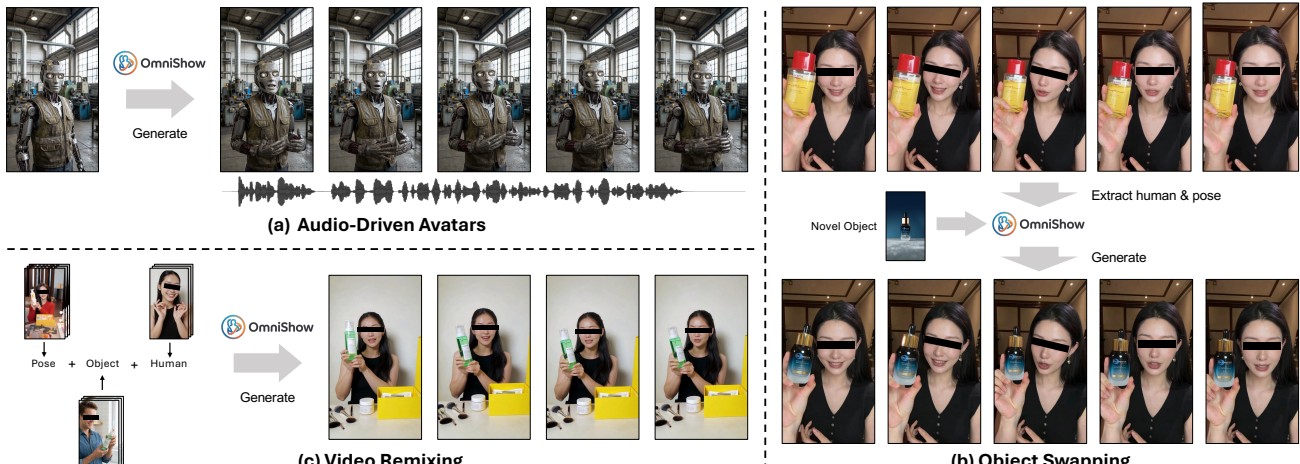

*Figure 8.* **OMNISHOW enables broader applications.** OMNISHOW unlocks creative potential such as (a) audio-driven avatars, (b) object swapping, and (c) video remixing, demonstrating robust compositional control and extensible capabilities.

*Table 3.* **Quantitative results of ablation studies.** We evaluate three key components of OMNISHOW, including (a) Unified Channel-wise Conditioning, (b) Gated Local-Context Attention, and (c) Decoupled-Then-Joint Training.

(a)

| Variant | FaceSim↑ | NexusScore↑ | AES↑ |
|---|---|---|---|
| Token-Concat | 0.601 | 0.344 | 0.466 |
| w/o Ref. Rec. Loss | 0.678 | 0.352 | 0.466 |
| Ours | **0.707** | **0.353** | **0.471** |

(b)

| Variant | Sync-C↑ | Sync-D↓ | AES↑ |
|---|---|---|---|
| w/o Audio Context | 8.872 | 7.878 | 0.533 |
| w/o Attn.-Map Constraints | 2.201 | 13.01 | **0.545** |
| w/o Adaptive Gating | 8.872 | 7.819 | 0.529 |
| Ours | **9.023** | **7.419** | 0.540 |

(c)

| Variant | NexusScore↑ | Sync-D↓ | AES↑ |
|---|---|---|---|
| Single-Stage (Only RA2V) | 0.345 | 13.11 | 0.453 |
| Multi-Stage (R2V → RA2V) | 0.360 | 13.23 | 0.473 |
| Multi-Stage (A2V → RA2V) | 0.342 | **7.38** | 0.456 |
| Ours | **0.364** | 8.14 | **0.474** |

ing 32 GPUs, as shown in Table 3c. Specifically, "Only RA2V" means training directly on RA2V data, while "R2V/A2V→RA2V" denotes training on R2V/A2V data and then RA2V data. These baselines fail to fully leverage heterogeneous data: single-stage training underperforms due to poor convergence, while naive multi-stage approaches struggle with incorporating new modal inputs. In contrast, our paradigm achieves the best trade-off between reference consistency and audio-visual synchronization.

**Effectiveness on the A2V Task.** As audio is arguably the most special modality to tame, we conduct an additional evaluation on the model derived from the A2V training, denoted as OMNISHOW-A2V. Benefiting from our Gated Local-Context Attention, this model achieves state-of-the-art performance on the EMTD benchmark (Meng et al., 2025), as illustrated in Table 2. Notably, its Sync-C score of 6.49 surpasses competitive methods like MultiTalk (Kong et al., 2025). These results validate the efficacy of the proposed technique on the A2V task, which also establishes a robust foundation for the final unified model.

**Broader Applications.** Beyond standard benchmarks, OMNISHOW exhibits remarkable versatility in broader applications, as illustrated in Figure 8. First, it naturally supports *audio-driven avatars* as a subset capability. Given a human reference image and an audio input, it effectively animates characters with synchronized speech and natural expressions. Moreover, OMNISHOW enables more complex

creative tasks by integrating into specific workflows. For instance, in *object swapping*, we can replace the original object in a video with a novel one while keeping the human subject and pose consistent. Furthermore, *video remixing* allows for synthesizing new videos by recombining pose, object, and human references from different sources. These results highlight its strong compositional control, paving the way for diverse real-world content creation.

## 5. Conclusion

In this work, we introduced **OMNISHOW**, a unified framework for HOIVG. By orchestrating text, reference image, audio, and pose conditions, OMNISHOW achieves precise multimodal control and high-quality video generation. Through the carefully designed architecture and training strategy, we effectively achieve harmonious multimodal injection and heterogeneous data utility. Extensive comparison and detailed ablation studies on HOIVG-Bench verified the superiority of OMNISHOW.

**Limitations and Future Work.** Despite the promising results, limitations remain. Current evaluation focuses on 5-second clips, and extreme motion, conflicting inputs, or AI-generated test samples may still cause mild bias or artifacts. Future work will include RL-based post-training, scaling data and model capacity, and supporting richer conditions such as camera trajectories or reference videos.

## Acknowledgments

This study was supported in part by the InnoHK initiative of the Innovation and Technology Commission of the Hong Kong Special Administrative Region Government via the Hong Kong Centre for Logistics Robotics, and in part by the Research Grants Council of the Hong Kong Special Administrative Region, China, under Project 14202125. We thank Qijun Gan, Pan Xie, and Yifu Zhang for helpful discussions, and Liyang Chen for his advice on baseline reproduction. We appreciate the support from Yuqi Zhang and Hao Yang regarding internal devkits, and the assistance from Ruibiao Lu, Chao Zhang, and Wei Feng on data collection.

## Impact Statement

While our primary goal is to push the boundaries of video generation research, we acknowledge the potential societal implications, including both positive impacts, such as enhancing accessibility in digital content creation and education, and risks, such as misuse for generating deceptive or harmful content. To proactively mitigate these concerns, we advocate for responsible use and the continuous development of safeguards to ensure this technology ultimately serves constructive and beneficial societal purposes.

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

# A. Training Data Collection

A video generation model's potential is mainly bounded by the richness, diversity, and scale of the data upon which it is trained. However, high-quality data for Human-Object Interaction Video Generation (HOIVG) is scarce, as it necessitates the simultaneous validity of text, reference images, audio, and pose conditions. To address this challenge, we construct a large-scale, heterogeneous dataset through a rigorous pipeline designed to meet specific sub-task standards. As illustrated in Figure 9, our pipeline consists of three main stages.

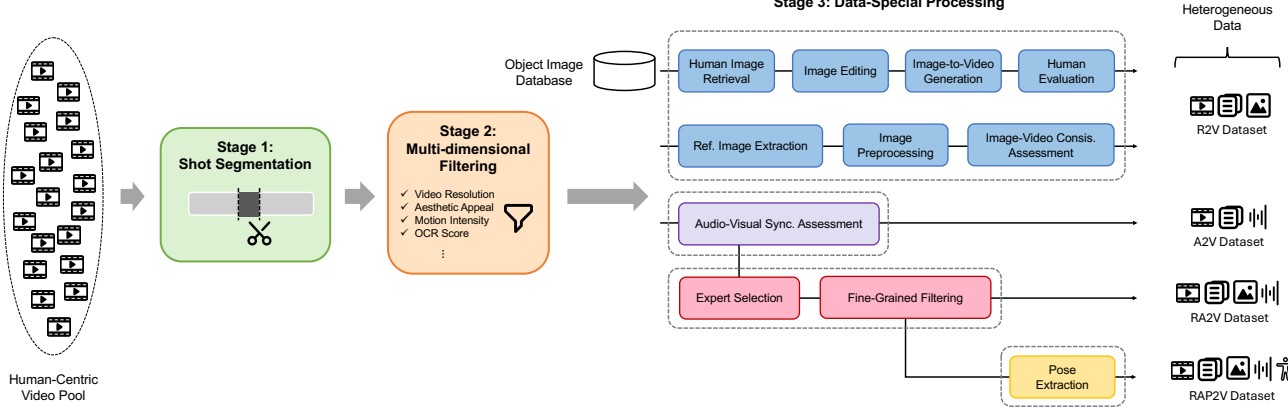

*Figure 9.* **Data collection pipeline.** We first segment videos into isolated shots and then filter them by diverse metric scores. Subsequently, specialized processing workflows are applied to construct multiple heterogeneous datasets.

**Stage 1: Shot Segmentation.** We begin by collecting a massive in-house human-centric video pool. To ensure temporal coherence and content focus, we apply a shot segmentation algorithm (Castellano, 2025) to decompose long videos into individual clips. This process isolates continuous shots, effectively removing scene transitions and ensuring that each clip contains a single, uninterrupted visual narrative.

**Stage 2: Multi-dimensional Filtering.** Following segmentation, we implement a filtering process to rigorously remove low-quality samples. We employ a multi-dimensional criteria set to screen the clips based on various metrics, such as (1) *video resolution*, ensuring high visual fidelity; (2) *aesthetic appeal*, filtering for visually pleasing composition and lighting; (3) *motion intensity*, selecting clips with sufficient movement to facilitate effective motion learning; and (4) *OCR score*, removing clips with excessive on-screen text or watermarks that might degrade generation quality. This comprehensive filtering significantly reduces noise and ensures that only high-quality candidates proceed to the next phase.

**Stage 3: Data-Special Processing.** To maximize data utilization and support various training objectives, we process the filtered clips into four distinct heterogeneous datasets, each tailored for specific multimodal condition training (for notational conciseness, we omit "text" from respective task names):

- **R2V (Reference-to-Video) Dataset:** For synthesized data, we utilize an internal object image database. The process involves retrieving relevant human images, performing image editing to compose human and object concepts, and generating corresponding videos via an internal image-to-video generation model. To ensure quality, we incorporate a human evaluation step, where crowd experts filter out samples exhibiting distortions or excessive "AI-ness" (*e.g.*, loss of fine details or over-smoothing). Additionally, we extract reference images from real videos, perform preprocessing such as image super-resolution, and conduct image-video consistency assessment to guarantee alignment.

- **A2V (Audio-to-Video) Dataset:** This dataset focuses on audio-driven generation. We perform audio-visual synchronization assessment to select clips where the audio (mainly speech) aligns perfectly with the visual actions, discarding samples with significant misalignment.

- **RA2V (Reference+Audio-to-Video) Dataset:** As a high-quality dataset for joint training, this dataset undergoes the most stringent curation. It combines the criteria of the previous datasets and adds an expert selection phase followed by fine-grained filtering. Each video clip is independently reviewed to ensure it meets the highest standards of visual quality and semantic consistency across both reference image and audio modalities.

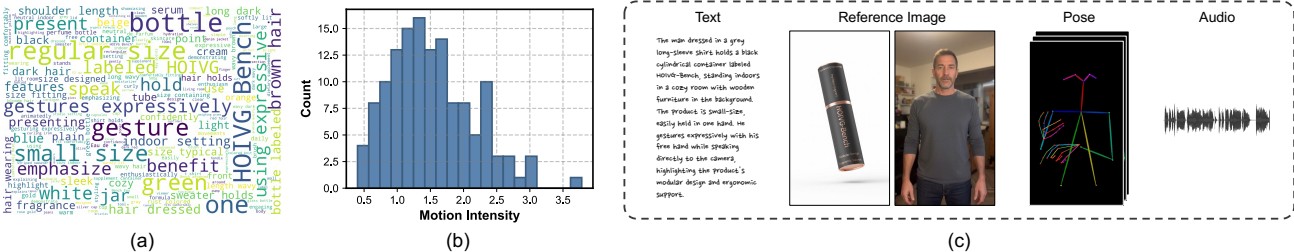

*Figure 10.* **Statistics and example of HOIVG-Bench.** (a) Word cloud of text prompts illustrating the variety of human-object interaction scenarios. (b) Motion intensity distribution of original videos calculated via dense optical flow (Farnebäck, 2003), reflecting a dynamic range of pose conditions. (c) A representative sample from the benchmark, featuring text, reference images, pose, and audio. Constructed through a rigorous pipeline, the benchmark provides high-quality data to evaluate the synergy of diverse multimodal inputs.

- **RAP2V (Reference+Audio+Pose-to-Video) Dataset:** Finally, for the final fine-tuning, we extend a high-quality subset of the RA2V dataset to apply pose extraction. We extract per-frame human pose skeletons from the validated videos, providing ground truth necessary for precise motion control.

By further integrating video captioning to provide comprehensive descriptions of human subjects, objects, actions, environments, and interaction details, this pipeline successfully curates a total of $\mathcal{O}(1m)$ **clips**, amounting to around **3500 hours**. This collection serves as a robust foundation for training our HOIVG model.

## B. Detailed HOIVG-Bench Construction

While the main paper provides a condensed overview of HOIVG-Bench, the benchmark itself is built through a dedicated multi-stage pipeline designed to ensure multimodal completeness, privacy compliance, and evaluation diversity. In this section, we describe the detailed sample construction process, including how we curate source videos, derive human and object references, synthesize aligned audio, and enforce manual quality control before finalizing the released benchmark.

**Sample Overview.** In HOIVG-Bench, each sample is equipped with a detailed textual caption, a human reference image, an object reference image, semantically aligned audio, and a coherent pose sequence.

**Construction Pipeline.** We adopt a rigorous data curation and processing pipeline to construct the benchmark:

- **Video Curation:** We initially select raw video from an in-house video library. The selection criteria include: (1) video duration exceeding 4 seconds; (2) the presence of clear human-object interactions; and (3) a diverse range of human attributes (*e.g.*, gender, age, ethnicity) and object categories (*e.g.*, daily necessities, tools) to ensure data diversity.

- **Object Image Acquisition:** To simulate real-world generation scenarios, we avoid simply cropping objects from the video. Instead, we utilize Nano Banana (Google, 2026) to modify the original objects' textures and colors while adding sufficient fine-grained details, resulting in high-quality reference object images.

- **Human Image Acquisition:** Considering privacy protection and identity de-identification, we generate reference human images based on video screenshots using Nano Banana. These generated images maintain stylistic similarity to the original subjects while altering identity features, ensuring compliance and testing generalization.

- **Pose Extraction:** We employ DWPose (Yang et al., 2023) to extract per-frame human pose skeletons from the original videos, serving as the ground truth signal for motion control.

- **Audio Synthesis:** To construct semantically consistent audio inputs, we design a two-stage generation process. First, GPT-4o (Hurst et al., 2024) is utilized to generate a speech script focused on describing the target object. Subsequently, GPT-4o analyzes the gender and age attributes of the human reference image, and ElevenLabs (ElevenLabs, 2026) is invoked to synthesize high-quality speech audio with matching timbres.

Through the meticulous construction process described above, we have curated a high-quality benchmark, with statistics and an example visualized in Figure 10. Note that the decision to use AI-generated human and object images was a carefully deliberated choice to comply with privacy, ethical, and legal guidelines for public release. Moreover, to mitigate potential domain bias, we have conducted manual checks to filter out images with noticeable "AI-ness," ensuring the benchmark closely reflects real-world data.

*Table 4.* **Quantitative comparison with the cascaded baseline on the RAP2V task.** We compare our end-to-end OMNISHOW with a representative cascaded baseline (VACE (Jiang et al., 2025)+LatentSync (Li et al., 2024)). Our approach outperforms the cascaded baseline across all evaluation metrics, highlighting the superiority of multi-condition unification.

| Method | Text Align. | Reference Consistency | | Audio-Visual Sync. | | Pose Accuracy | | Video Quality | | | |
|---|---|---|---|---|---|---|---|---|---|---|---|
| | TA↑ | FaceSim↑ | NexusScore↑ | Sync-C↑ | Sync-D↓ | AKD↓ | PCK↑ | AES↑ | IQA↑ | VQ↑ | MQ↑ |
| Cascaded Baseline | 6.885 | 0.591 | 0.341 | 7.016 | 7.823 | 0.198 | 0.340 | 0.417 | 0.709 | 10.05 | 3.911 |
| OMNISHOW (Ours) | **7.134** | **0.645** | **0.353** | **7.699** | **7.674** | **0.172** | **0.478** | **0.424** | **0.725** | **11.06** | **5.880** |

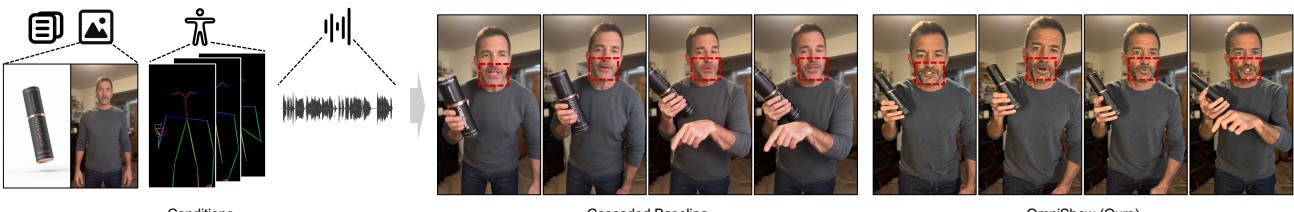

Conditions        Cascaded Baseline        OmniShow (Ours)

*Figure 11.* **Qualitative comparison with the cascaded baseline on the RAP2V task.** Compared to the cascaded baseline (VACE (Jiang et al., 2025)+LatentSync (Li et al., 2024)), our OMNISHOW generates visually coherent videos with precise lip synchronization and superior visual quality. Notably, it avoids the artifacts and blur often introduced by isolated lip-synchronization generation, particularly under complex occlusions. *Zoom in for better view.*

## C. More Implementation Details

Our model is built upon the 12B Waver 1.0 model (Zhang et al., 2025), which utilizes the Multimodal Diffusion Transformer (MMDiT) architecture (Esser et al., 2024). The training framework employs Flow Matching (Lipman et al., 2022) to supervise video sequences. The entire training process involves an initial phase using 480p videos followed by a high-resolution phase using 720p videos. To manage the computational demands of long video training samples (up to 241 frames at 24 fps), we set the Ulysses-style sequence parallelism (Jacobs et al., 2023) size to 8 within our distributed setup. Furthermore, to maximize training efficiency, all features are extracted offline before training. Notably, incorporating the proposed Gated Local-Context Attention increases the model scale by merely 0.3B parameters. This efficiency is attributed to the insights provided by the adaptive gating mechanism, as shown in Figure 4. In summary, our approach achieves high-quality video generation while maintaining a highly efficient parameter footprint relative to the 12B backbone.

## D. Additional Comparison

To further validate the effectiveness of our end-to-end multi-condition unification, we construct a cascaded baseline for the RAP2V task. Specifically, this baseline sequentially combines VACE (Jiang et al., 2025) (for RP2V generation) with LatentSync (Li et al., 2024) (for audio-driven lip-synchronization). We believe it serves as a representative approach to sequentially synthesize RAP2V samples.

As shown in the quantitative results in Table 4, the cascaded baseline struggles across multiple key metrics. In contrast, our OMNISHOW achieves clearly superior performance across the board. Notably, our end-to-end approach yields markedly better audio-visual synchronization (*e.g.*, a Sync-C of 7.699 compared to 7.016) and pose accuracy (*e.g.*, a PCK of 0.478 compared to 0.340). Furthermore, video and motion quality metrics are also substantially higher overall (such as a VQ of 11.06 versus 10.05), highlighting the effectiveness of simultaneously modeling all conditions. Furthermore, we provide qualitative comparisons in Figure 11. OMNISHOW consistently generates highly coherent videos with better visual fidelity compared to the cascaded baseline. Specifically, conducting isolated lip-synchronization generation as a post-processing step in the cascaded baseline leads to visual artifacts and blur, especially under mouth occlusions, which can be attributed to the inherent limitations of non-end-to-end approaches. This comparison strongly highlights the necessity and advantages of our unified end-to-end framework for HOIVG.

*Table 5.* **Quantitative results of additional ablation studies.** We further evaluate two design choices within the proposed techniques, including (a) different RoPE strategies for positional encoding of the pseudo-frames in Decoupled-Then-Joint Training and (b) The impact of context window size in Gated Local-Context Attention.

(a)

| RoPE Strategy | FaceSim↑ | NexusScore↑ | AES↑ |
|---|---|---|---|
| Temporal Shift | 0.675 | 0.351 | 0.468 |
| Spatiotemporal Shift | 0.279 | 0.339 | 0.456 |
| Native (Ours) | **0.707** | **0.353** | **0.471** |

(b)

| Context Setup | Sync-C↑ | Sync-D↓ | AES↑ |
|---|---|---|---|
| Context Window = 1 | 8.872 | 7.878 | 0.533 |
| Context Window = 11 | 7.020 | 9.588 | 0.527 |
| Context Window = 5 (Ours) | **9.023** | **7.419** | **0.540** |

# E. Additional Ablation Studies

## E.1. Position Embeddings for Unified Channel-wise Conditioning

Modern MMDiT models use 3D Spatiotemporal Rotary Positional Embeddings (RoPE) (Su et al., 2024). As for the RoPE setup of pseudo-frames, we adopt the native strategy, treating pseudo-frames and original video frames as a continuous sequence starting from $T = 0$. We compared this against a "Temporal Shift" strategy (using negative indices like $T = -1, -2$) and a "Spatiotemporal Shift" strategy (adding spatial offsets) following (Hu et al., 2025). As shown in Table 5a, empirical results in the R2V task favor our native strategy. We attribute this to the model's pre-training on standard continuous video. Artificial shifts create mismatches for the channel-wise conditioning, whereas our approach aligns with the model's inherent expectation of temporal continuity, allowing it to effectively utilize the reference context.

## E.2. Window Size of Audio Context Packing

The proposed Gated Local-Context Attention employs a sliding window of $w = 5$ to capture phonetic context. We validated this against $w = 1$ (*i.e.*, no audio context) and $w = 11$. As illustrated in Table 5b, quantitative results show $w = 5$ offers the best synchronization. A small window ($w = 1$) fails to capture temporal correlation, resulting in jittery transitions between phonemes and overreaction. Conversely, an excessive window ($w = 11$) causes "over-smoothing," where broad context dilutes the fine-grained cues needed for precise lip synchronization. Thus, $w = 5$ strikes the optimal balance, providing sufficient context without obscuring the instantaneous audio signal required for precise synchronization.

# F. More Qualitative Results

To provide a substantially more comprehensive and detailed assessment of the generative capabilities of OMNISHOW, we present an extensive set of additional qualitative results in Figure 12, Figure 13, and Figure 14. These visualization results cover a highly diverse and challenging range of multimodal conditioning settings, including R2V, RA2V, RP2V, and RAP2V.

Across all of these varied and demanding scenarios, our proposed model consistently exhibits superior generation quality, which is characterized by high-fidelity reference identity preservation, natural motion dynamics, and precise audio-visual synchronization. Most notably, the provided additional samples further serve to vividly illustrate the model's exceptional robustness in effectively handling complex spatial human-object interactions as well as large pose variations, without compromising underlying visual details. Furthermore, we showcase the unique capability of our unified framework by presenting compelling results under the fully combined conditioning setting (*i.e.*, RAP2V), conclusively verifying its profound effectiveness in seamlessly orchestrating all input modalities simultaneously to generate highly coherent, temporally smooth, and richly expressive videos. Video demonstrations are available on our project page.

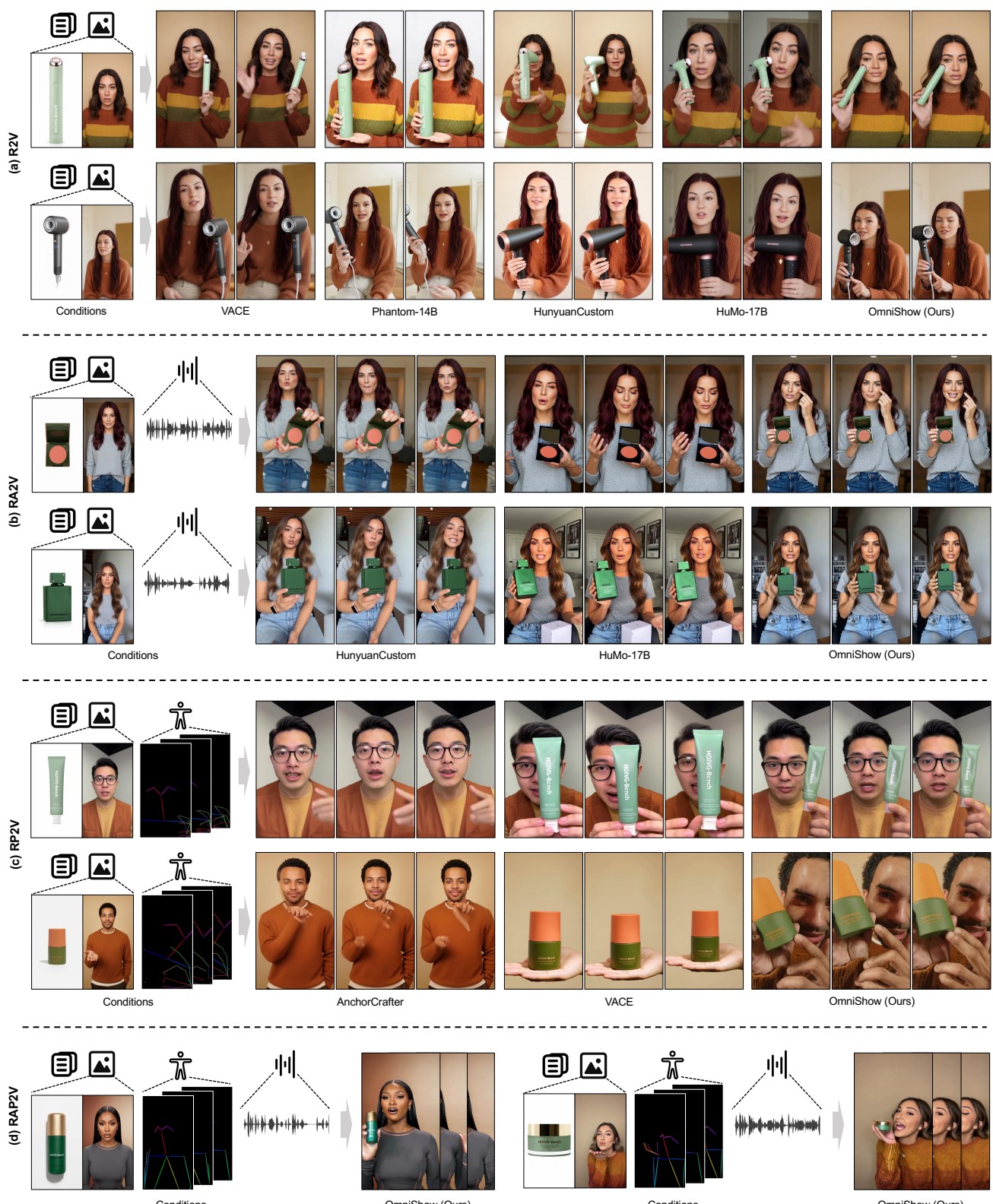

*Figure 12.* **More extensive and diverse qualitative results.** We present additional generated results from our OMNISHOW and other methods across various multimodal condition settings, showing the state-of-the-art performance of OMNISHOW. Note that OMNISHOW is the first-of-its-kind framework supporting the full spectrum of four multimodal inputs required by HOIVG. *We recommend visiting our project page for video demonstrations.*

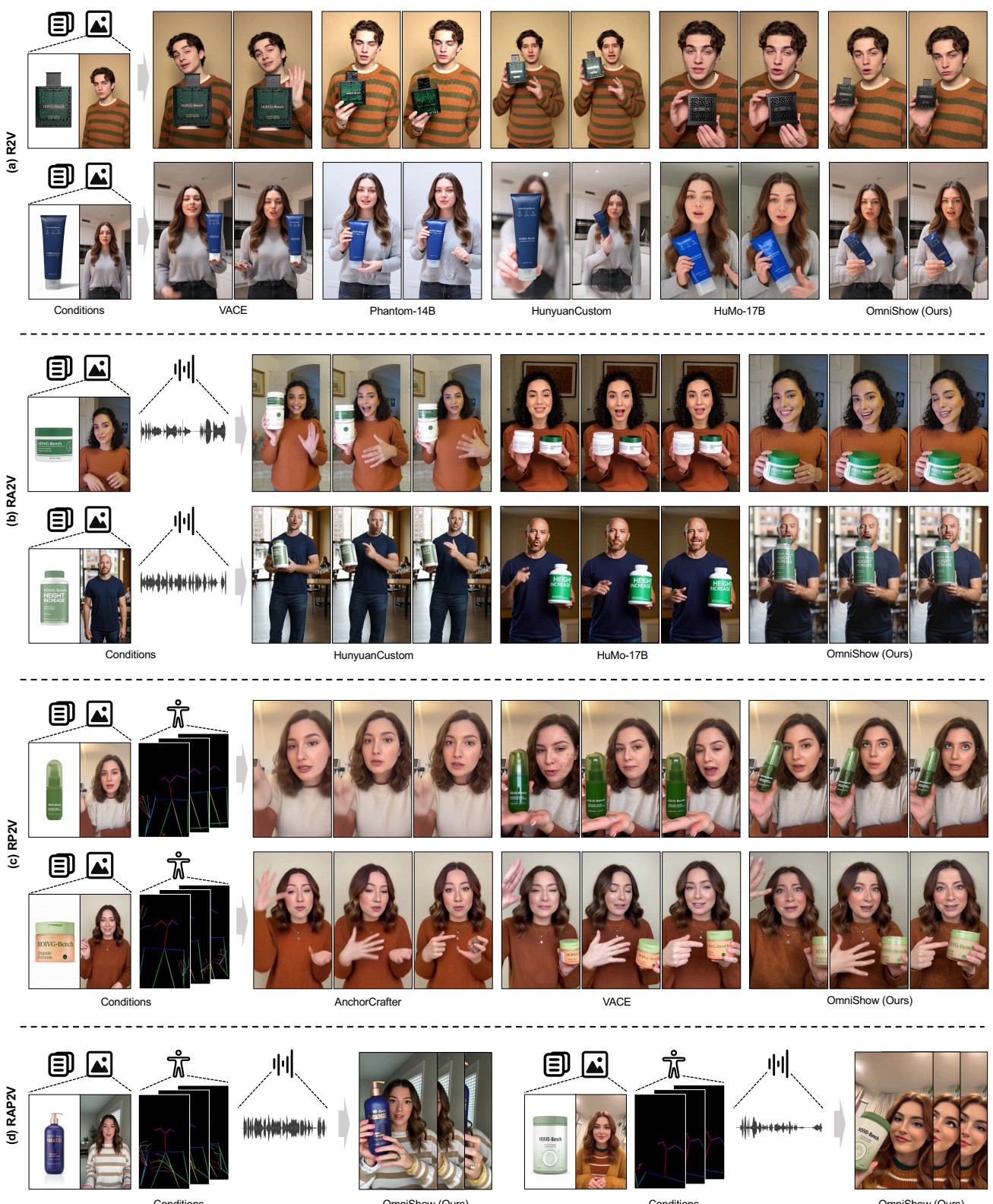

*Figure 13.* **More extensive and diverse qualitative results.** We present additional generated results from our OMNISHOW and other methods across various multimodal condition settings, showing the state-of-the-art performance of OMNISHOW. Note that OMNISHOW is the first-of-its-kind framework supporting the full spectrum of four multimodal inputs required by HOIVG. *We recommend visiting our project page for video demonstrations.*

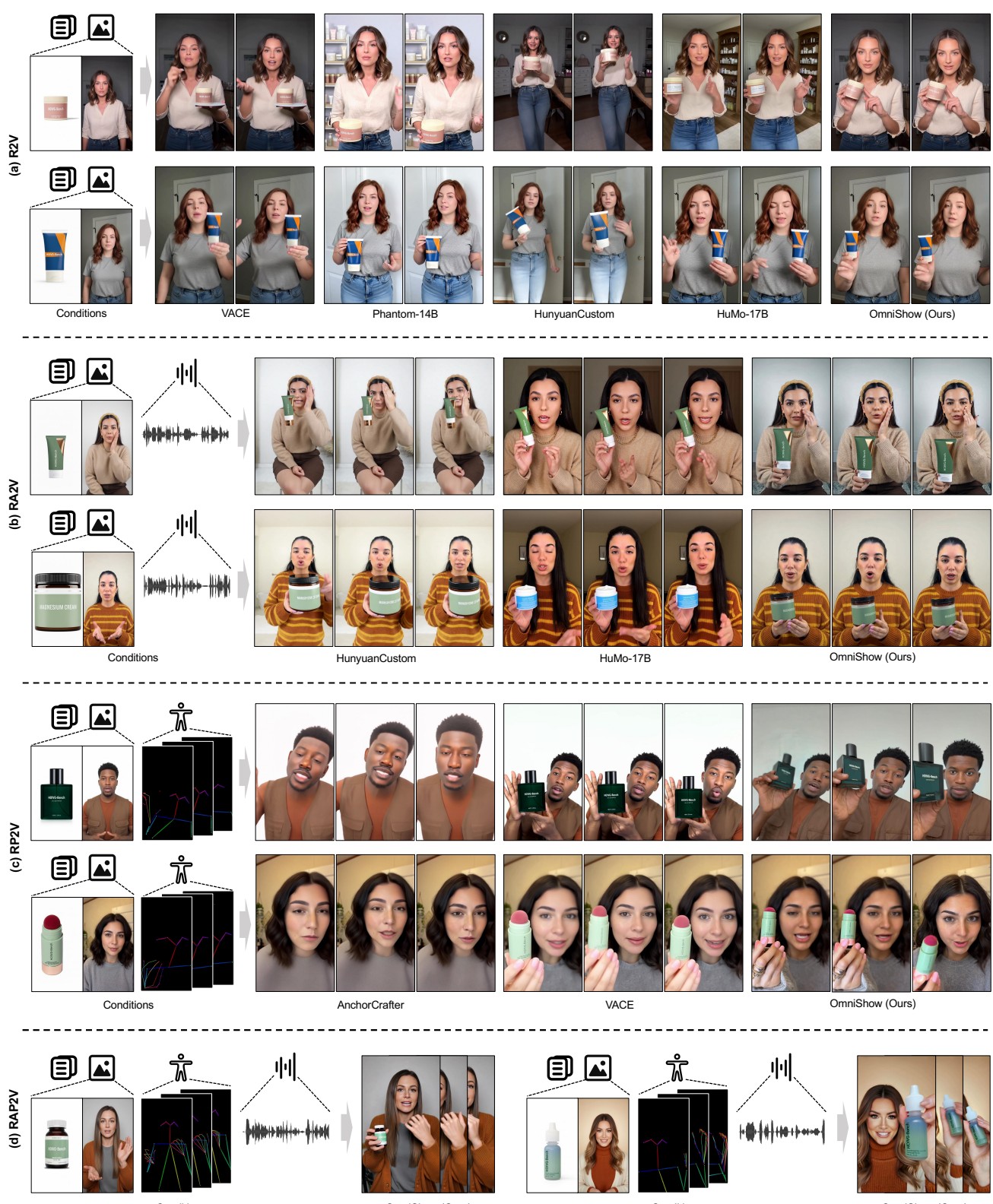

*Figure 14.* **More extensive and diverse qualitative results.** We present additional generated results from our OMNISHOW and other methods across various multimodal condition settings, showing the state-of-the-art performance of OMNISHOW. Note that OMNISHOW is the first-of-its-kind framework supporting the full spectrum of four multimodal inputs required by HOIVG. *We recommend visiting our project page for video demonstrations.*

