# OpenReview forum: "OmniShow: Unifying Multimodal Conditions for Human-Object Interaction Video Generation"
_ICML.cc/2026/Conference — ICML 2026 regular_

### Official Review · Reviewer_FsQQ · 2026-03-09

**Soundness:** 3
**Presentation:** 3
**Significance:** 3
**Originality:** 2
**Overall Recommendation:** 3
**Confidence:** 4

**Summary:**

This paper introduces OmniShow, a novel framework for Human-Object Interaction Video Generation (HOIVG) that  simultaneously orchestrate four modalities: text, reference images, audio, and pose. To tackle the challenges of architectural integration and data scarcity, the authors propose (1) Unified Channel-wise Conditioning for efficient injection of visual and pose signals; (2) Gated Local-Context Attention for precise audio-visual synchronization with adaptive gating; and (3) a Decoupled-Then-Joint training strategy to leverage heterogeneous sub-task datasets (R2V, A2V) for unified multimodal control. Experiments demonstrate that OmniShow achieves state-of-the-art or competitive performance across various multimodal settings.

**Compliance With Llm Reviewing Policy:**

Affirmed.

**Final Justification:**

I maintain rejection as the second-round rebuttal still not convince me of their design novelty, for example, the training from specialists to a generalist is a conventional training scheme, not a tailored design in such multi-modal conditioned generation tasks.

**Key Questions For Authors:**

Reproducibility: Given that HOIVG-Bench are built on proprietary data and closed-source models, how can the research community reproduce your results or fairly compare their methods against OmniShow? Will the authors release the generated object/human images and synthesized audio for the benchmark?

Architectural Generalization: The techniques are validated on Waver 1.0. How dependent are these designs on this specific architecture? Would they transfer effectively to other popular video diffusion backbones ?

Analysis of Model Merging: The weight interpolation between R2V and A2V models appears somewhat heuristic. Is there a principled reason for this specific ratio?

Utility of Pose in the Final Stage: The paper mentions that pose is introduced only in the final fine-tuning stage to prevent overfitting. Does the model retain its ability to generate high-quality videos when the pose signal is partially or fully occluded or noisy at inference time?

**Limitations:**

Yes

**Strengths And Weaknesses:**

Strength:
The paper presents the first end-to-end framework capable of handling all four modalities (text, reference image, audio, pose) for HOIVG, and the proposed solutions are reasonable yet effective.


Weakness:
1. While the unified framework is novel, the individual technical components are largely adaptations of existing ideas. Channel-wise concatenation is common in controllable generation. Masked attention and gating mechanisms are well-established techniques in sequence modeling and model merging.

2. The construction of HOIVG-Bench relies heavily on proprietary data and models for object modification (Nano Banana), human image generation (Nano Banana), and audio synthesis (GPT-4o, ElevenLabs). This reliance severely limits the reproducibility of the benchmark by the wider research community and raises questions about its long-term availability and accessibility. A benchmark built on such a foundation is difficult for others to use for direct comparison.

3. The framework is built upon the 12B-parameter Waver 1.0 model. It is unclear how much of the final performance is attributable to the proposed conditioning techniques versus the inherent power of this large, pre-trained backbone. An ablation study starting from a smaller, open-source backbone would better isolate the contribution of OmniShow's specific architectural designs.

---

> ### Author Rebuttal · Authors · 2026-03-31
>
> We sincerely appreciate the time you dedicated to reviewing our paper and the insightful feedback you provided. In the following, we present our point-by-point responses to your comments.
>
> ---
>
> > **Q1:** Component Technical Contributions
>
> While some underlying concepts have existed, we have made **targeted and effective improvements**, integrating them into a unified framework. Below, we clarify the technical contributions of the mentioned components:
>
> 1. **Channel Concatenation:** Beyond simple concatenation, we additionally extended **pseudo-frames** for the noisy video to expand its conditioning capacity, coupling them with **Reference Reconstruction Loss** to explicitly compel the model to perceive and retain semantic details. This technique injects both reference images and pose within a unified paradigm, *minimizing the task adaptation gap and achieving efficient conditioning* (see ablation in *Table 3(a)*).
>
> 2. **Masked Attention:** We designed the **audio context packing** strategy to aggregate contextual audio features from raw audio signals, then applied **audio-visual temporal alignment** via attention map constraints. This synergistic design explicitly prevents video frames from "seeing" irrelevant audio segments while still retaining essential context, thereby *achieving precise audio-visual synchronization* (see ablation in *Table 3(b)* and *Table 4(b)*).
>
> 3. **Adaptive Gating:** We employ near-zero initialized gating vectors to gradually absorb audio features, avoiding severe distribution shifts. This design **ensures stable training** and ultimately *yields better performance* (see ablation in *Table 3(b)*). Moreover, we leverage these learnable vectors as **explicit analytical indicators**. The empirical observation (see *Figure 4*) guides us to strategically insert audio modules only into dual-stream blocks, thereby *optimizing parameter efficiency* (see the *response to Reviewer RuE4 (Q1)* for more details).
>
> ---
>
> > **Q2:** Reproducibility of HOIVG-Bench
>
> We commit to completely open-sourcing our HOIVG-Bench. The release will include:
> 1. **All test samples** including paired text, reference images, audio, and pose inputs.
> 2. **Full evaluation code** for launching evaluations.
> 3. **Comprehensive documentation** detailing environment setup, execution instructions, and usage tutorials.
>
> The community will be able to conduct fair and direct comparisons without relying on any commercial APIs.
>
> ---
>
> > **Q3:** Model Capacity
>
> Please see *the response to Reviewer oxny (Q2)*, clarifying that among the 10B-scale models that meet industry-grade quality standards, our model achieves **superior overall performance** (see the *response to Reviewer 8yWx (Q1)* for further video comparison) while being the **most parameter-efficient**  (**ours 12.3B vs. others 13B~17B**).
>
> Regarding the model capacity for ablation studies, experimenting on the final 10B-scale backbone is more representative, as it directly reflects ultimate model capabilities. Within this setting, the various proposed techniques have been rigorously isolated and comprehensively validated (see *Section 4.3*).
>
> ---
>
> > **Q4:** Architectural Generalization
>
> The proposed techniques are highly generalizable to standard DiT architectures, as evidenced by their *non-intrusive* nature:
> 1. **Unified Channel-wise Conditioning** only requires the backbone to natively support channel concatenation (e.g., Wan2.1-I2V).
> 2. **Gated Local-Context Attention** is designed to function as a plug-and-play module for base models.
> 3. **Decoupled-Then-Joint Training** is inherently a model-agnostic training strategy.
>
> Therefore, applying these techniques to other backbones stands as a promising avenue. Given limited time and computational resources during the rebuttal phase, we leave this migration as a valuable direction for future work.
>
> ---
>
> > **Q5:** Rationale for Model Merging Ratio
>
> Since R2V and A2V target distinct modality inputs, deriving a closed-form analytical solution for the merging ratio is mathematically intractable. The ratio selection is guided by a *principled observation*: audio synchronization (relying on *fine-grained temporal alignment*) is significantly more sensitive to weight disruption than visual identity (relying on *global appearance features*). Thus, the A2V model requires a higher weight of 0.6 to preserve lip-sync capabilities, while 0.4 is sufficient for the R2V model to maintain reference consistency.
>
> ---
>
> > **Q6:** Robustness to Pose Conditions
>
> We tested our model with corrupted pose sequences (e.g., missing, occluded, or noisy) in RAP2V and RP2V. The results ([*Video*](https://anonymous-5483.github.io/pose_robustness.mp4)) show that our model exhibits *satisfactory robustness*. Specifically, our training strategy effectively prevents overfitting to pose, enabling it to tolerate imperfect pose inputs while generating high-quality, temporally coherent videos.

---

> > ### Author Rebuttal · Reviewer_FsQQ · 2026-04-04
> >
> > While the authors addressed some of my concerns, I still have following concerns:
> >
> > 1. I agree with Reviewer oxny that the benefit from backbone scale is not verified. No controlled experiment regarding backbone is provided. It remains unclear whether gains come from the 12B backbone or the proposed mechanisms.
> >
> > 2. The authors still not convince me of their design novelty, the components reads as an engineering integration rather than a methodological breakthrough.
> >
> > 3. Qualitative results show blurred logos and trademarks, contradicting claims of commercial applicability.

---

> > > ### Author Response · Authors · 2026-04-08
> > >
> > > We deeply appreciate your continued engagement, and would be very grateful if you could consider our responses below in your final justification.
> > >
> > > ---
> > > > **MQ1:** Fairness of Baseline Comparison
> > >
> > > ***New Backbone-Aligned Experiment on a Smaller Backbone***
> > >
> > > We applied our techniques to *Wan2.1-1.3B* and compared the model against HuMo-1.7B, which is also built upon the 1.3B backbone. We designed this experiment for two goals: (i) to verify the superiority of our designs under a backbone-aligned setting; and (ii) to validate whether they generalize well to a smaller backbone.
> > >
> > > Quantitative results (*[Table](https://anonymous-5483.github.io/1_3B_comp.png)*) show that *our method still outperforms HuMo across most metrics*. The results of our method were achieved with only a rough set of hyperparameters, indicating potential for further improvement.
> > >
> > > This direct comparison clearly shows that our performance gains stem from our method designs, rather than relying on the underlying model capacity.
> > >
> > > ***Clarification on the Original Experimental Setup***
> > >
> > > Moreover, we would also like to provide some context to help understand the considerations behind our original setup regarding backbones.
> > >
> > > First, given the diverse preferences in model selection across different works and the prohibitive cost of re-training video generation models, forcing all methods to align on an identical backbone is practically infeasible.
> > >
> > > Therefore, the video generation community has widely adopted a more pragmatic evaluation paradigm: *methods with different backbones are compared under their own best-performing configurations in main experiments, followed by ablation studies under a fixed backbone to validate the effectiveness of the proposed mechanisms.*
> > >
> > > As shown in (*[Table](https://anonymous-5483.github.io/recent_papers.png)*), a broad range of recent top-tier papers have adopted this cross-backbone evaluation protocol, indicating that it has been widely accepted and has become a de facto common practice.
> > >
> > > ---
> > > > **MQ2:** Clarification on Novelty
> > >
> > > We respectfully argue that our model is not a naive *"engineering integration"* but a *principle-driven architecture* to address the fundamental challenges of HOIVG. We would like to highlight two key points behind the method details:
> > >
> > > 1. **Philosophy of Minimalist Intervention:** By deeply understanding the underlying input structure and the learning dynamic of DiTs, we explored the least disruptive injection methods. Specifically, we proposed *Unified Channel-wise Conditioning* to repurpose the native channel-concat mechanism, and introduced *Gated Local-Context Attention* to insert efficient attention modules guided by adaptive gating vectors, achieving multimodal control capabilities with minimal architectural changes.
> > >
> > > 2. **Training from Specialists to a Generalist:** Rather than merely training on scarce paired data, we propose *Decoupled-Then-Joint Training* to fully leverage heterogeneous data from diverse sub-tasks. Crucially, by merging R2V and A2V models, the resulting model exhibits striking emergent RA2V capabilities, and then be jointly trained on RA2V and RAP2V. This finding, that controllability can emerge via weight merging, is also an inspiring insight that we hope will motivate further exploration.
> > >
> > > Ultimately, our model achieves robust multimodal controllability without compromising generation quality. We believe that in the era of large-scale foundation models, not only pure *"methodological breakthroughs"* merit publication, but also *profound insights and pragmatic method co-designs* are valuable to be shared with the community.
> > >
> > > ---
> > > > **MQ3:** Blur in Qualitative Results
> > >
> > > ***Clarification on Our Claims***
> > >
> > > We must respectfully clarify that **we did not claim our model has achieved "commercial applicability"**.
> > >
> > > The closest descriptions we found are: (i) "while ..., real-world applications, such as ..." (Lines 35-40); and (ii) "..., paving the way for diverse real-world content creation" (Lines 422-423).
> > >
> > > These statements aim to highlight the value of HOIVG and the potential of OmniShow, rather than claiming a flawless product.
> > >
> > > ***The Blur Issue***
> > >
> > > Preserving fine details remains a challenging open problem for controllable video generation. Specifically, the occasional blur could stem from two practical factors:
> > > 1. **Inherent detail loss during VAE compression and reconstruction**, which particularly affects small patterns and typography;
> > > 2. **Overly intense human motion**, which naturally introduces motion blur similar to what occurs in real-world camera captures.
> > >
> > > We acknowledge that the blur issue is not fully solved and there is still ample room for improvement. In future work, we plan to further alleviate this issue by *jointly optimizing the VAE and diffusion model*, as well as *exploring RL-based post-training methods*.

---

### Official Review · Reviewer_8yWx · 2026-03-09

**Soundness:** 3
**Presentation:** 3
**Significance:** 3
**Originality:** 4
**Overall Recommendation:** 5
**Confidence:** 4

**Summary:**

In their paper, the authors proposed OmniShow, a framework for Human-Object Interaction Video Generation (HOIVG). This was achieved by introducing three elements: Unified Channel-wise Conditioning for efficiently injecting image and pose cues, Gated Local-Context Attention to ensure precise audio-visual synchronization, and Decoupled-Then-Joint Training strategy to effectively harness data. Given the scarcity of high-quality data for HOIVG, the authors also introduced HOIVG-Bench with five key metrics to evaluate the performance of their proposed framework against other similar solutions in three different configurations: R2V, RA2V, and RP2V. Although no other solution accepts all the modalities that OmniShow framework accepts, it outperformed others on the EMTD benchmark. Finally, ablation studies on the three main elements demonstrate their effectiveness and robustness.

**Compliance With Llm Reviewing Policy:**

Affirmed.

**Final Justification:**

I believe authors have addressed my concerns and questions properly and provided sound further presentation materials to back their previous claims. Based on the rebuttal I changed my evaluation from Weak Accept to Accept.

**Key Questions For Authors:**

1. Since this is a video generation paper, the lack of supplementary videos makes it impossible to verify claims about smooth motion and precise lip-sync. Providing a link to an anonymous project page or video files showing stable temporal dynamics will significantly raise my score for soundness and presentation.

2. The evaluations only use 5-second clips. How well does OmniShow handle longer videos? Showing that the model maintains audio-visual sync without error accumulation over longer periods will improve my score for the work's significance.

3. The benchmark uses AI-generated images for the human references. Does this give the model an unfair advantage compared to using real-world photos? Clarifying that the model works just as well with noisy, real-world images will increase my confidence in the benchmark's validity.

**Limitations:**

The authors should add a dedicated "Limitations" section before the Conclusion. This should explicitly cover:

1. The limitation of only evaluating short, 5-second clips and the risk of temporal drift in longer videos.
2. The potential bias of using AI-generated human reference images in their benchmark rather than real photos .
3. Known failure cases, such as how the model handles severe occlusions or complex overlapping interactions.

**Strengths And Weaknesses:**

Strengths:

[Significance] OmniShow is the first framework which can simultaneously orchestrate text, reference images, audio, and pose conditions for Human-Object Interaction Video Generation (HOIVG), addressing a key step for practical digital content creation.

[Originality] The Decoupled-Then-Joint Training strategy is a practical and original solution to the scarcity of paired, multi-condition datasets. Training specialized R2V and A2V models first, and then fusing them via weight interpolation, effectively utilizes fragmented, heterogeneous data.

[Soundness] The Unified Channel-wise Conditioning efficiently injects image and pose cues by augmenting noisy video tokens with pseudo-frames. This mechanism preserves the native input structure and token distribution of the base model, minimizing adaptation and maintaining pre-trained generative priors.

[Presentation] HOIVG-Bench, an evaluation resource for multi-modal HOI tasks, offers a foundation for future benchmarking. Curated with over 100 samples, detailed captions, human/object references, aligned audio, and extracted pose sequences, it provides a valuable resource for the field.

Weaknesses:

[Soundness / Presentation] Although the authors provided frame samples from the output of the model, since the model is generating videos and the flow and consistency of frames is a pivotal attribute of the performance, it would have been better to have some samples from the proposed model and other models in the supplemental materials for a better qualitative understanding. For example, claims regarding "natural body movements," "precise lip synchronization," and the avoidance of "frozen body" issues cannot be robustly verified by reviewers through static image grids alone.

[Soundness] For the HOIVG-Bench, the authors utilized an image generation model to create the human reference images to ensure privacy. While practical for privacy, using AI-generated images as conditioning inputs for an evaluation benchmark may introduce inherent biases that fail to accurately reflect the model's performance on the real world.

[Soundness] The human preference study is statistically weak, relying on only 16 evaluators assessing a randomly selected subset of just 20 samples and only in the AR2V setting. Given the subjective nature of this assessment, a larger-scale and more diverse user study is necessary.

[Significance] The quantitative metrics and qualitative analyses on the HOIVG-Bench are strictly standardized to 5-second video clips. It is mentioned that this has been done for having a fair comparison, but it remains unproven how the proposed Gated Local-Context Attention and temporal dynamics scale, or if they suffer from significant drift over longer, more complex interactions.

---

> ### Author Rebuttal · Authors · 2026-03-31
>
> We sincerely appreciate the time you dedicated to reviewing our paper and the insightful feedback you provided. In the following, we present our point-by-point responses to your comments.
>
> ---
>
> > **Q1:** Video Comparison
>
> We completely agree that video samples are pivotal for qualitative evaluation. As you suggested, we provide comprehensive video comparison in ([*Video*](https://anonymous-5483.github.io/main_comp.mp4)), covering the *R2V, RA2V, RP2V, and RAP2V* settings.
>
> The visual results clearly demonstrate that our method consistently produces *natural body movements* and *precise audio-visual alignment* with *smooth temporal dynamics*. Most importantly, OmniShow is the **first** unified framework capable of *handling all these multimodal conditions simultaneously*.
>
> ---
>
> > **Q2:** AI-Generated Human Images in HOIVG-Bench
>
>
> The decision to use AI-generated human images was a carefully deliberated choice to comply with privacy, ethical, and legal guidelines for public release.
>
> Moreover, to mitigate potential domain bias, we have conducted *manual checks* to filter out images with noticeable "AI-ness" during benchmark construction, ensuring the benchmark closely reflects real-world data.
>
> To alleviate concerns regarding real-world generalization, we tested our model using *real-world human photos*. The results ([*Video*](https://anonymous-5483.github.io/real_world_images.mp4)) show that OmniShow still exhibits *strong robustness*, proving its capabilities are not inflated by or limited to AI-generated human images.
>
> ---
>
> > **Q3:** Scope of User Studies
>
> As you suggested, we have expanded the scope of user studies. Specifically, we expanded the *RA2V user study* to include *30* evaluators, and conducted a completely new *RP2V user study* with *33* evaluators. In doing so, we also ensured a diverse demographic pool among the participants.
>
> Considering that excessive samples could reduce participant engagement, we maintained the assessment at 20 samples, which is considered a reasonable volume for the user studies of video generation.
>
> The expanded user studies ([*Figure*](https://anonymous-5483.github.io/expanded_user_studies.png)) corroborate our initial findings, showing that OmniShow is *consistently preferred by human evaluators* in both *condition adherence* and *visual quality*.
>
> ---
>
> > **Q4:** Performance on Longer Videos
>
> We present *up-to-10-second* RA2V results ([*Video*](https://anonymous-5483.github.io/long_videos.mp4)) to demonstrate OmniShow's capabilities beyond standard-length clips.
>
> The results show that our *Gated Local-Context Attention* also works well when generating longer videos. The model consistently maintains *precise audio-visual synchronization* and *coherent temporal dynamics* throughout the extended duration.
>
> ---
>
> > **Q5:** Adding a "Limitations" Section
>
> We sincerely thank you for this highly constructive suggestion. We will add a dedicated "Limitations" section in our final version. This section will explicitly and thoroughly discuss the aspects you kindly mentioned.

---

> > ### Author Rebuttal · Reviewer_8yWx · 2026-04-01
> >
> > Thanks to the authors for providing the responses. My concerns are addressed and I will increase my score.

---

> > > ### Author Response · Authors · 2026-04-02
> > >
> > > Dear Reviewer 8yWx,
> > >
> > > Thank you for your acknowledgement and for increasing your score to 5.
> > >
> > > We would like to express our sincere gratitude to you, as your helpful feedback and valuable suggestions have been instrumental in elevating the overall quality of our work.
> > >
> > > We will incorporate the relevant revisions, analyses, and results into the main paper or appendix.
> > >
> > > We are truly grateful for your time and effort, and we wish you all the best in your future research endeavors.
> > >
> > > Best regards,
> > >
> > > Authors of Paper 5483

---

### Official Review · Reviewer_oxny · 2026-03-11

**Soundness:** 2
**Presentation:** 2
**Significance:** 2
**Originality:** 2
**Overall Recommendation:** 3
**Confidence:** 3

**Summary:**

The paper introduces an end-to-end framework, ie OmniShow, for Human-Object Interaction Video Generation, which aims to simultaneously orchestrate four distinct conditions: text, reference image, audio, and pose. Built on a 12B parameter MMDiT backbone, the framework utilizes a Unified Channel-wise Conditioning mechanism to inject image and pose cues. It leverages a Gated Local-Context Attention for audio-visual synchronization and proposes a Decoupled-Then-Joint Training strategy to leverage fragmented sub-task datasets. The authors also evaluate their approach on a proposed benchmark, HOIVG-Bench. The results demonstrate competitive performance across various multi-modal settings.

**Compliance With Llm Reviewing Policy:**

Affirmed.

**Final Justification:**

I maintain my recommendation for rejection because the rebuttal failed to resolve core concerns regarding the fairness of baseline comparisons and the concrete mechanisms behind the four-condition unification. Specifically, the authors did not adequately disentangle their architectural contributions from the massive 12B backbone capacity, nor did they provide necessary evidence.

**Key Questions For Authors:**

- Could the aiuthors explicitly define "pseudo-frames" earlier in the text to improve readability?

- Are the ablation models trained to full convergence on the 8 GPUs, and how does their total training compute (e.g., effective batch size and steps) compare to the main model trained on 128 GPUs?

- How would OmniShow compare quantitatively and qualitatively against a cascaded pipeline of existing SOTA models for the full RAP2V task?

- Since the adaptive gating vector initializes at 1e-5, I am curious about how many training steps does it take for the audio conditioning to actually start influencing the generation process?

**Limitations:**

yes

**Strengths And Weaknesses:**

Strengths:

1.The proposed method tackles the unification of four modalities (text, image, audio, pose) simultaneously, which is a highly ambitious and relevant problem for human-centric video generation.

2.The introduction of the HOIVG-Bench benchmark with carefully curated samples involving diverse humans and daily objects is a positive contribution to the community.

3.The qualitative results for complex interactions, particularly in the Reference-Audio-Pose-to-Video (RAP2V) setting, show impressive visual fidelity and hand-contact accuracy.





Weakness

1.The term "pseudo-frames" appears early in the text (Section 1) without a clear definition. What exactly are pseudo-frames in this context, and how do they function mechanically before the reader reaches Section 3.2? This makes the introductory explanation difficult to parse.

2.The core motivation of the paper is unclear and not targeted. Stating that the framework's capability is underpinned by "integrating efficient and pragmatic techniques with the synergistic utilization of heterogeneous training data" is far too general to serve as the motivation for a specific paper.

3.The third listed contribution, "Extensive experiments on our OmniShow-Bench and other benchmarks validate that OMNISHOW achieves state-of-the-art performance...", is a statement of fact regarding the evaluation outcomes, not a distinct methodological or theoretical contribution.

4.There is a formatting error in Table 1 regarding the IQA metric. The best and second-best results are incorrectly marked (bolded and underlined).

5.The baseline comparisons lack strict fairness regarding model capacity. OmniShow is built upon Waver 1.0, a massive 12B parameter backbone. Baselines like AnchorCrafter or VACE are not contextualized by parameter count in the main text. Since performance obviously grows with larger parameter counts (Table 1), it is difficult to disentangle whether the performance gains come from the proposed architectural mechanisms or simply the sheer capacity of the 12B backbone.

6.The ablation studies are conducted in a significantly mismatched compute environment. The main model is trained on a massive cluster of 128 GPUs , but the ablations are performed on just 8 GPUs. It is entirely unclear if these ablated models were trained to full convergence or if this drastic reduction in compute artificially penalizes the ablated variants.

7.Because the paper notes that no existing methods simultaneously support all four conditions, there is no direct baseline comparison for the full RAP2V task. The authors should have constructed a naive "cascaded" baseline to properly validate the necessity and superiority of their end-to-end framework.

---

> ### Author Rebuttal · Authors · 2026-03-31
>
> We sincerely appreciate the time you dedicated to reviewing our paper and the insightful feedback you provided. In the following, we present our point-by-point responses to your comments.
>
> ---
>
> > **Q1:** Writing Issues
>
> We will make the following revisions:
>
> 1. **Pseudo-frames:** Mechanically, they are additional tokens concatenated along the temporal dimension of noisy video tokens, organized frame-by-frame, hence the term "pseudo-frames". They serve as expansion "slots", allowing the model to inject reference image tokens when original "slots" are fully occupied by pose conditions. We will explicitly define "pseudo-frames" early in *Section 1*.
>
> 2. **Motivations:** In this statement, "Efficient and pragmatic techniques" refers to *Unified Channel-wise Conditioning* and *Gated Local-Context Attention*, designed to resolve *Challenge (i)*. "Synergistic utilization" refers to *Decoupled-Then-Joint Training*, designed to overcome *Challenge (ii)*. We will refine this to clearly target the challenges of HOIVG.
>
> 3. **Contribution 3:** We will rewrite this contribution to focus on HOIVG-Bench rather than emphasizing our performance.
>
> 4. **Formatting Error:** We will correct the formatting error of the IQA metric.
>
> Thank you for pointing these out.
>
> ---
>
> > **Q2:** Model Capacity
>
> We fully acknowledge that 10B-scale models generally outperform 1B-scale models, which is a consensus in the community. To further clarify, we list model parameter counts as follows:
>
> |HuMo|Phantom|VACE|HunyuanVideo|OmniShow|AnchorCrafter|
> |:-:|:-:|:-:|:-:|:-:|:-:|
> |17B/1.7B|14B/1.3B|14B|13B|**12.3B**|1.5B|
>
> As 1B-scale models cannot meet industry-grade quality, practical applications typically focus on 10B-scale models, where **OmniShow is the smallest and most parameter-efficient**. Despite its size, it achieves *overall SoTA performance across various multi-modal settings*, outperforming larger models like HuMo, HunyuanVideo, Phantom, and VACE (see the *response to Reviewer 8yWx (Q1)* for further video comparison).
>
> Moreover, OmniShow is the **first** unified framework capable of *simultaneously handling all four conditions*, which is a unique capability absent in other baselines.
>
> ---
>
> > **Q3:** Clarification of Ablation Studies
>
> First, we must clarify that **in ablation studies, our method was also trained on the 8-GPU setup**, ensuring a fair comparison with other variants under identical training configurations.
>
> The detailed comparisons between the 8-GPU ablation and 128-GPU main training setups are as follows:
> 1. **Effective Batch Size:** The 128/8-GPU setups used effective batch sizes of 32/2 at 480p. This is the *only* hyperparameter difference.
> 2. **Training Steps:** To compensate for the smaller batch size, we properly increased total training steps in proportion to the 8-GPU setup, allowing models to traverse a similar number of samples as the formal training to guarantee *full convergence*. All variants (including ours) were trained with the same number of steps.
>
> Regarding the rationale, conducting initial explorations and ablations on a scaled-down compute setup is a standard, resource-efficient practice. Given a fixed compute budget, an 8-GPU setup allowed for testing more schemes, while performing such comprehensive ablations on 128 GPUs would be prohibitively expensive.
>
> ---
>
> > **Q4:** RAP2V Comparison with a Cascaded Baseline
>
> Following your suggestion, we constructed a cascaded baseline by combining VACE (for RP2V generation) with LatentSync [Li et al.] (for audio-driven lip-sync). We believe this serves as a representative approach to sequentially synthesize RAP2V samples.
>
> The qualitative results are included in ([*Video*](https://anonymous-5483.github.io/cascaded_comp.mp4)), and the quantitative results are reported below:
>
> | Method | TA↑ | FaceSim↑ | NexusScore↑ | Sync-C↑ | Sync-D↓ | AKD↓ | PCK↑ | AES↑ | IQA↑ | VQ↑ | MQ↑ |
> |:-|:-:|:-:|:-:|:-:|:-:|:-:|:-:|:---:|:---:|:---:|:---:|
> | Cascaded | 6.885 | 0.591 | 0.341 | 7.016 | 7.823 | 0.198 | 0.340 | 0.417 | 0.709 | 10.05 | 3.911 |
> | **Ours** | **7.134** | **0.645** | **0.353** | **7.699** | **7.674** | **0.172** | **0.478** | **0.424** | **0.725** | **11.06** | **5.880** |
>
> Overall, the cascaded baseline struggles across multiple metrics. Notably, conducting isolated lip-sync leads to *severe artifacts* (especially under *mouth occlusions*), attributed to its *non-end-to-end* nature.
>
> In contrast, our OmniShow achieves superior performance, maintaining *precise lip-sync* and *stable pose following*. This comparison highlights the value of an *end-to-end* framework for multi-condition unification of HOIVG.
>
> ---
>
> > **Q5:** Audio Influence Steps
>
> As shown in ([*Video*](https://anonymous-5483.github.io/audio_steps.mp4)), the model first learns to synchronize body movements with the audio rhythm (10k steps), then gradually aligns lip dynamics (20k steps). As training continues, the overall audio-visual alignment progressively improves and stabilizes (30-40k steps).

---

> > ### Author Rebuttal · Reviewer_oxny · 2026-04-05
> >
> > Thanks for the rebuttal. The rebuttal addresses several presentation-level concerns and partially clarifies the ablation setting and cascaded comparison. However, my main concerns regarding fairness of baseline comparison (Merely listing parameter counts is not enough to disentangle the effect of architectural design from backbone scale.), the concrete mechanism of four-condition unification (the rebuttal claims OmniShow is the first unified framework, but does not clearly explain how the four modalities are jointly represented and fused in a unified manner), and the practical efficiency of the proposed end-to-end design (The paper does not provide evidence regarding computational efficiency, such as training/inference cost, latency, FLOPs, or memory usage, especially compared with cascaded alternatives.) remain insufficiently resolved. In addition, key metrics, supported conditions, and model sizes should be summarized in one table for transparent comparison.

---

> > > ### Author Response · Authors · 2026-04-08
> > >
> > > We deeply appreciate your continued engagement, and would be very grateful if you could consider our responses below in your final justification.
> > >
> > > ---
> > > > **MQ1:** Fairness of Baseline Comparison
> > >
> > > We sincerely appreciate your suggestion and have conducted a new experiment to directly address your core concern regarding backbone scale, followed by a clarification of our original experimental rationale.
> > >
> > > ***New Backbone-Aligned Experiment on a Smaller Backbone***
> > >
> > > To further verify the effectiveness of our techniques, we applied them to *Wan2.1-1.3B* and compared the model against HuMo-1.7B, which is also built upon the 1.3B backbone. We designed this experiment for two goals: (i) to demonstrate the superiority of our designs under a strictly backbone-aligned setting; and (ii) to validate whether our proposed techniques generalize well to a smaller backbone.
> > >
> > > As shown by the quantitative results on RA2V (*[Table](https://anonymous-5483.github.io/1_3B_comp.png)*), *our method still outperforms HuMo across most metrics*. The results of our method were achieved with only a rough set of hyperparameters due to the time constraints of rebuttal, indicating potential for further improvement.
> > >
> > > By isolating the proposed techniques from the original backbone, this direct comparison clearly validates that our performance gains stem fundamentally from our method designs, rather than relying on the underlying model capacity.
> > >
> > > ***Clarification on the Original Experimental Setup***
> > >
> > > While the above experiment rigorously validates our method's contribution, we would also like to provide some context to help understand the considerations behind our original experimental setup regarding backbones.
> > >
> > > First, given the diverse preferences in model selection across different works and the prohibitive cost of re-training video generation models, forcing all methods to align on an identical backbone is practically infeasible.
> > >
> > > Therefore, the video generation community has widely adopted a more pragmatic evaluation paradigm: *methods with different backbones are compared under their own best-performing configurations in main experiments, followed by ablation studies under a fixed backbone to validate the effectiveness of the proposed mechanisms.*
> > >
> > > As shown in (*[Table](https://anonymous-5483.github.io/recent_papers.png)*), a broad range of recent top-tier papers have adopted this cross-backbone evaluation protocol, indicating that it has been widely accepted and has become a de facto common practice.
> > >
> > > ---
> > > > **MQ2:** Concrete Mechanism of Condition Unification
> > >
> > > We clarify how the four conditions are represented and fused:
> > >
> > > 1. **Text:** Text tokens are extracted via the text encoder and injected through the original mechanism (token-concat with noisy video tokens before the first block) for subsequent self-attention with noisy video tokens.
> > > 2. **Ref. Image & Pose:** Reference images and pose video are mapped to image and video tokens via the VAE encoder. Then, they are integrated into noisy video tokens by *Unified Channel-wise Conditioning*.
> > > 3. **Audio:** Raw audio signals are pre-processed and projected into audio tokens, which interact with noisy video tokens via cross-attention within *Gated Local-Context Attention*.
> > >
> > > In this way, all modalities are respectively transformed into tokens and jointly modeled within a single MMDiT architecture.
> > >
> > > ---
> > > > **MQ3:** Efficiency of Ours vs. Cascaded
> > >
> > > A cascaded alternative for RAP2V (e.g., VACE+LatentSync) requires sequentially running multiple models, leading to repeated VAE encoding/decoding, multiple independent denoising processes, and severe I/O overhead. In contrast, our OmniShow completes multi-condition generation in an end-to-end manner.
> > >
> > > We report several intuitive efficiency metrics on an H100 GPU for generating a 5-second RAP2V video (24 FPS, 720P):
> > >
> > > |Method|Total Size↓|Peak VRAM↓|Latency↓|
> > > |:-|:-:|:-:|:-:|
> > > |Cascaded|14B+1B|**63.7GB**|465.17s|
> > > |**Ours**|**12.3B**|73.9GB|**293.57s**|
> > >
> > > Our base model adopts the MMDiT architecture, where more intermediate activation variables lead to a higher peak VRAM. However, *our end-to-end design still significantly reduces inference latency by ~37%* via avoiding redundant executions, proving its better practical efficiency.
> > >
> > > ---
> > > > **MQ4:** Summary Table
> > >
> > > As requested, we summarize the supported conditions and sizes of all comparison models in one table:
> > > |Method|Size|Text|Ref. Image|Audio|Pose|
> > > |:-|:-|:-:|:-:|:-:|:-:|
> > > |Phantom|14B/1.3B|✓|✓|-|-|
> > > |HunyuanCustom|13B|✓|✓|✓|-|
> > > |HuMo|17B/1.7B|✓|✓|✓|-|
> > > |AnchorCrafter|1.5B|✓|✓|-|✓|
> > > |VACE|14B|✓|✓|-|✓|
> > > |**OmniShow (Ours)**|12.3B|✓|✓|✓|✓|
> > >
> > > Since our experiments involved multiple settings (i.e., R2V, RA2V, and AR2V), aggregating all key metrics into a single table would be overly complex and confusing. Therefore, please refer to *Table 1* for a metric comparison.

---

### Official Review · Reviewer_RuE4 · 2026-03-13

**Soundness:** 4
**Presentation:** 4
**Significance:** 4
**Originality:** 3
**Overall Recommendation:** 6
**Confidence:** 4

**Summary:**

The authors propose a framework named OmniShow for human-object interaction video generation (HOIVG). They introduce a unified channel-wise conditioning mechanism to inject pose and reference image cues into the video latents. They also propose a gated local-context attention module that aligns audio features with human dynamics via masked attention to facilitate temporal alignment. Additionally, they develop a decoupled-then-joint training strategy to effectively leverage heterogeneous data. Finally, they introduce a HOIVG benchmark and demonstrate state-of-the-art performance on both this benchmark and existing ones.

**Compliance With Llm Reviewing Policy:**

Affirmed.

**Final Justification:**

I already provided the highest rating and I wish to maintain it.

**Key Questions For Authors:**

Within my area of expertise, I have no outstanding questions. The authors have addressed everything I could think of in the paper. This is a strong submission that appears technically sound. That said, the following questions arise out of my curiosity:
1. Why do the authors believe that audio features require a special adaptive gating mechanism for conditioning? What other features could leverage this design choice?
2. How would the conditioning framework extend to other modalities, such as depth maps or 3D priors? Which modalities could be incorporated directly without disrupting the generative priors, and which would require more careful injection? What informed the authors’ design decisions in this regard?

**Limitations:**

Yes

**Strengths And Weaknesses:**

### Strengths
- The paper is well-written and easy to follow. The authors provide sufficient motivation for each of the proposed components.
- The novel components introduced are adequately supplemented with ablation experiments that validate their individual contributions.
- This work represents a meaningful step forward in multimodal conditional video generation.

---

> ### Author Rebuttal · Authors · 2026-03-31
>
> We sincerely appreciate the time you dedicated to reviewing our paper and the insightful feedback you provided. In the following, we present our point-by-point responses to your comments.
>
> ---
>
> > **Q1:** Discussion on Adaptive Gating
>
> ***Why Audio Features Require Adaptive Gating***
>
> We employ adaptive gating for audio feature injection based on two primary considerations:
>
> **1. Better Training Stability:**
>
> During early training, the newly initialized audio attention modules are insufficiently trained, meaning audio features remain *poorly aligned* with video features. Directly injecting these unaligned features without our adaptive gating mechanism would cause severe feature distribution shifts, leading to *training instability*.
>
> By initializing adaptive gating vectors to *near-zero values*, we allow the model to gradually absorb audio features as training progresses. This design facilitates *stable training* and thus brings **superior performance**, which is empirically validated by our ablation study in *Table 3(b)*.
>
> **2. Architectural Design Indicator:**
>
> Beyond stabilization, learnable gating vectors also serve as an *explicit indicator* that guides our architectural design, specifically, determining the *optimal insertion locations* for audio attention modules.
>
> As illustrated in *Figure 4*, our empirical analysis reveals that the norm of the gating vectors is *notably larger* in dual-stream blocks (indicating a significant role) while remaining *negligible* in single-stream blocks. We also provide a curve showing the variation of the average gating vector norm over training steps ([*Figure*](https://anonymous-5483.github.io/norm_curve.png)) as further evidence.
>
> This observation provides clear guidance: we should insert audio attention modules *only* into the dual-stream blocks. This strategic placement helps to *enhance parameter efficiency*. Specifically, our method adds only **0.3B** parameters (a mere **2.5%** increase over the 12B base model, totaling **12.3B**). In contrast, existing methods like HuMo introduce **3B** additional parameters (an approximately **21.4%** increase over their 14B base model, totaling **17B**).
>
> ***What Other Features Could Leverage This Design***
>
> Beyond audio, this design is versatile and can be extended to a wide range of auxiliary features that require injection via a newly initialized module. This "soft-start" approach effectively ensures a smooth and stable feature alignment process.
>
> ---
>
> > **Q2:** Discussion on Conditioning Paradigms
>
> ***Extension to Other Modalities***
>
> For modalities that can be directly rendered into videos via operations like color mapping (e.g., depth maps or normal maps), we can encode them using the pretrained VAE encoder, mapping them into the exact same latent space as the native video features.
> Consequently, they can be naturally integrated using our channel-wise concatenation strategy. In fact, this is exactly how we process pose conditions (rendering them into skeleton videos).
>
> For non-renderable modalities (e.g., 3D point clouds or camera trajectory coordinates), which are highly abstract and structurally heterogeneous, direct injection is unfeasible. It is more principled to employ specialized encoders followed by attention mechanisms (similar to our audio attention modules) to effectively facilitate their alignment and interaction with video features.
>
> ***Direct Incorporation vs. Careful Injection***
>
> As discussed above, modalities renderable into the video space are more suitable for direct incorporation, as they can share the same latent distribution as the target videos.
>
> Non-renderable modalities, however, exhibit significant domain gaps and require careful injection strategies to prevent the disruption of generative priors.
>
> ***Design Decisions***
>
> Our design decisions are fundamentally grounded in the core principle of *effectively preserving the base model's native capabilities*.
>
> Accordingly, for both reference images and poses (renderable), we integrate them via our *Unified Channel-wise Conditioning*, fully exploiting the shared video latent space. For audio (non-renderable), we selectively inject these features via our *Gated Local-Context Attention* into effective blocks, avoiding excessive architectural additions.
>
> We believe this principled strategy is the key factor in *OmniShow's successful balance across multimodal controllability and visual quality*.

---

> > ### Author Rebuttal · Reviewer_RuE4 · 2026-04-01
> >
> > Thanks to the authors for responding to my questions. I have already given the highest rating.

---

> > > ### Author Response · Authors · 2026-04-02
> > >
> > > Dear Reviewer RuE4,
> > >
> > > Thank you for your acknowledgement and for your highest rating.
> > >
> > > We sincerely appreciate your constructive comments and insightful questions regarding the details of our methodology, which have been very helpful in improving the quality and clarity of our work.
> > >
> > > We hope these discussions and analyses will provide valuable insights and benefit the broader research community.
> > >
> > > We are truly grateful for your time and effort, and we wish you all the best in your future research endeavors.
> > >
> > > Best regards,
> > >
> > > Authors of Paper 5483

---

### Decision · Program_Chairs · 2026-04-30

**Decision:**

Accept (regular)

**Comment:**

Reviewers agreed that OmniShow addresses the challenging problem of multimodal human-object interaction video generation and introduces a novel end-to-end framework with well-motivated components. While concerns were raised about backbone scale confounding, benchmark reproducibility, and incremental novelty, the authors convincingly addressed these in the rebuttal by providing backbone-aligned experiments on a smaller 1.3B model, committing to full open-sourcing of HOIVG-Bench, demonstrating superiority over a cascaded baseline, and releasing extensive video results and expanded user studies. The paper is therefore recommended for acceptance.